# Integrated genetic and epigenetic analysis of myxofibrosarcoma

Koichi Ogura[1,2,3], Fumie Hosoda[1], Yasuhito Arai[1], Hiromi Nakamura[1], Natsuko Hama[1], Yasushi Totoki[1], Akihiko Yoshida[4], Momoko Nagai[1], Mamoru Kato[1], Erika Arakawa[1], Wakako Mukai[1], Hirofumi Rokutan[1], Akira Kawai[2], Sakae Tanaka[3] & Tatsuhiro Shibata [1,5]

Myxofibrosarcoma (MFS) is a common adult soft tissue sarcoma characterized by an infiltrative growth pattern and a high local recurrence rate. Here we report the genetic and epigenetic landscape of MFS based on the results of whole-exome sequencing ($N = 41$), RNA sequencing ($N = 29$), and methylation analysis ($N = 41$), using 41 MFSs as a discovery set, and subsequent targeted sequencing of 140 genes in the entire cohort of 99 MFSs and 17 MFSs' data from TCGA. Fourteen driver genes are identified, including potentially actionable therapeutic targets seen in 37% of cases. There are frequent alterations in p53 signaling (51%) and cell cycle checkpoint genes (43%). Other conceivably actionable driver genes including *ATRX*, *JAK1*, *NF1*, *NTRK1*, and novel oncogenic *BRAF* fusion gene are identified. Methylation patterns cluster into three subtypes associated with unique combinations of driver mutations, clinical outcomes, and immune cell compositions. Our results provide a valuable genomic resource to enable the design of precision medicine for MFS.

[1] Division of Cancer Genomics, National Cancer Center Research Institute, 5-1-1 Tsukiji, Chuo-ku, Tokyo 104-0045, Japan. [2] Department of Musculoskeletal Oncology, National Cancer Center Hospital, 5-1-1 Tsukiji, Chuo-ku, Tokyo 104-0045, Japan. [3] Department of Orthopaedic Surgery, The University of Tokyo, 7-3-1 Hongo, Bunkyo-ku, Tokyo 113-8677, Japan. [4] Department of Pathology and Clinical Laboratories, National Cancer Center Hospital, 5-1-1 Tsukiji, Chuo-ku, Tokyo 104-0045, Japan. [5] Laboratory of Molecular Medicine, Human Genome Center, The Institute of Medical Science, The University of Tokyo, 4-6-1, Shirokanedai, Minato-ku, Tokyo 108-0071, Japan. Correspondence and requests for materials should be addressed to T.S. (email: tashibat@ncc.go.jp)

Soft tissue sarcomas are a diverse group of tumors originating from mesenchymal cells, which are generally classified according to their histological similarities to normally differentiated tissues. Current understanding of the key genomic aberrations in soft tissue sarcomas is limited to demonstration that some exhibit recurrent single genetic alterations, such as chromosomal translocations resulting in gene fusions (SS18-SSX in synovial sarcoma, FUS-DDIT3 in myxoid/round cell liposarcoma, or NAB2-STAT6 in solitary fibrous tumor)[1–3], or point mutations (KIT in gastrointestinal stromal tumors)[3,4].

Recently, comprehensive sequencing efforts have been undertaken, revealing the genomic landscapes in several sarcoma subtypes, including alterations of PRC2 in malignant peripheral nerve sheath tumor or MYOD1 in spindle cell/sclerosing rhabdomyosarcoma[4,5]; however, insufficient comprehensive sequencing has been performed in the majority of soft tissue sarcoma subtypes, particularly those exhibiting complex karyotypes, which are rare and comprise more than 50 histological subtypes[6].

Myxofibrosarcoma (MFS) was originally classified as a myxoid-type malignant fibrous histiocytoma and was reclassified as a distinct entity in the WHO classification of 2002 because of its characteristic biological behavior and clinical features, including an infiltrative growth pattern and a high propensity for persistent local recurrence. MFS is now recognized as a common adult soft tissue sarcoma, which particularly affects the elderly population. MFS tumors generally harbor highly complex karyotypes, sharing many of the aberrations observed in leiomyosarcoma (LMS) and undifferentiated pleomorphic sarcoma (UPS)[3]. Amplifications of chromosome 5p (rapamycin-insensitive binding partner of mTOR, CDH9, LIFR) and 1p/1q (PI4KB, ETV3, MCL1), and deletions of tumor-suppressor genes including CDKN2A/B and TP53, have been reported in MFS, as have loss-of-function mutations in NF1 (5/35) and PTEN (1/35)[7]. However, the molecular pathogenesis of MFS remains incompletely understood because the small number of cases analyzed to date were not investigated using integrated genetic and epigenetic analyses, and because of the recent change in the diagnostic definition of this disease.

We therefore conducted a large-scale integrated genetic and epigenetic study of 99 MFSs, pathologically confirmed according to the latest WHO classification system[6], and analyzed the molecular data along with those of an additional 17 MFS cases obtained from The Cancer Genome Atlas (TCGA) data set[8–10]. Here, based on these analyses, we identified recurrent driver genes, including novel BRAF fusion gene, and novel methylation clusters associated with unique combinations of driver mutations, clinical outcomes, and immune cell compositions.

## Results

### The landscape of genetic alterations in MFS.
We conducted WES of 41 paired tumor and normal samples. All tumors were obtained from the primary site. The mean coverage of WES was 131×, and 95% of coding regions were analyzed at a depth of more than 20×. A total of 1971 non-silent somatic mutations (mean, 48 mutations per case) were detected, consisting of 1676 missense, 116 nonsense, 42 splice site, 136 indel, and 8 read-through mutations. The median numbers of substitutions and indels per case were 38 (interquartile range, 32–45) and 1 (interquartile range, 0–3), respectively. A particularly high number of non-silent somatic mutations (349) were observed in a single case (MFS_01).

By evaluating point mutations alone, WES identified 127 recurrently mutated genes with $p < 0.05$ and top 50 genes are listed in Supplementary Data 1a. The gene with a significant $q$ value ($q < 0.1$) was TP53 only ($q = 1.89 \times 10^{-12}$). Recurrently mutated genes included not only known mutational targets in MFS, such as TP53 (9/41, 22%; $p = 9.42 \times 10^{-17}$), NTRK1 (3/41, 7%; $p = 0.001$), and NF1 (3/41, 7%; $p = 0.008$), but also previously unreported genes including ATRX (4/41, 10%; $p = 0.001$) and TET2 (3/41, 7%; $p = 0.032$).

Using GISTIC 2.0 analysis, significant focal amplifications ($q < 0.25$) and copy number losses (homozygous and heterozygous deletions) ($q < 0.01$) were identified at 29 loci (15 amplifications and 14 copy number losses), including known oncogenes (JAK1, 1p31.3; VGLL3, 3p11.2; CCND1, 11q13.2; SYK, 9q21.2; FOXA1, 14q11.2; NKX2-1, 14q11.2; KRAS, 12q12; and CDK6, 7q21.3) (Supplementary Fig. 1a and Supplementary Data 2a), and tumor suppressors (TP53, 17p13.1; MST1R, 3p21.31; CDKN2A and CDKN2B, 9p21.3; RB1, 13q14.2; and CDH1, 16p11.2) (Supplementary Fig. 1b and Supplementary Data 2b). Consideration of cumulative genetic alterations (point mutations, CNAs, and fusion genes) revealed that a total of 14 genes were identified as significantly altered: TP53, CDKN2B, CCND1, CDKN2A, KRAS, WNT11, NTRK1, MDM2, CDK6, GNAS, FOXA1, NKX2-1, SYK, and JAK1 (geometric mean q value <0.1; Supplementary Data 1b).

We subsequently performed targeted sequencing validation study of 140 selected genes and their promoter regions, including those identified by WES and associated genes, and those identified by a review of the relevant literature (Supplementary Table 1)[7,11], using samples from the entire cohort of 99 MFS cases, including the 41 discovery cases. The mean sequencing coverage across the targeted loci was 187×, with 96.7% achieving coverage >20×. Using this approach, we identified 252 non-silent somatic mutations in 98 genes. By adding data from 17 MFSs from TCGA database to these 99 MFSs (WES and targeted sequencing ($N = 41$), targeted sequencing ($N = 58$)), a total of 116 patients were investigated. The incidence and distribution of driver mutations and copy number alterations are presented in Fig. 1. Genes frequently mutated in the 116 MFS cases were TP53 (53/116, 46%), RB1 (21/116, 18%), CDKN2A (18/116, 16%), CDKN2B (18/116, 16%), NF1 (13/116, 11%), and NTRK1 (11/116, 9%). These alterations affected potential therapeutic targets (NF1, NTRK1, ATRX, CCND1, CDK6, NKX2-1, and JAK1) with a total frequency of 39% (45/116).

We compared mutation frequencies of these driver genes among the sample status: primary ($N = 96$), recurrent ($N = 17$), and metastatic ($N = 3$). None was statistically more frequent in metastatic samples. However, frequencies of GNAS (primary 0.0%, metastatic 0.0% vs. recurrent 17.6%; $p < 0.001$, $\chi^2$-test) and SETD2 (primary 1.0%, metastatic 0.0% vs. recurrent 11.8%; $p = 0.036$, $\chi^2$-test) mutations were significantly higher in recurrent specimens.

### Alterations in p53 signaling/cell cycle G1/S checkpoint.
We identified frequent alterations in genes related to p53 signaling (51%), along with those associated with the cell cycle checkpoint (43%), including RB1, CDKN2A/CDKN2B, CCND1, and CDK6 (Fig. 1). Overall, 66% (76/116) of MFSs harbored alterations in these pathways (Fig. 2a). Of note, alterations in RB1, CDKN2A/CDKN2B, and CCND1 were almost mutually exclusively, as were those in TP53 and MDM2 (Fig. 1). These findings indicate central roles for dysregulation of p53 signaling and G1/S cell cycle in the development of MFS.

### Other pathways recurrently affected in MFS.
Other recurrent mutational targets included genes encoding components of the receptor tyrosine kinase (RTK)-RAS-PI3K cancer pathway (31%) (Figs. 1 and 2b). In this pathway, amplification or hotspot mutations of KRAS and PIK3CA were identified, that have not

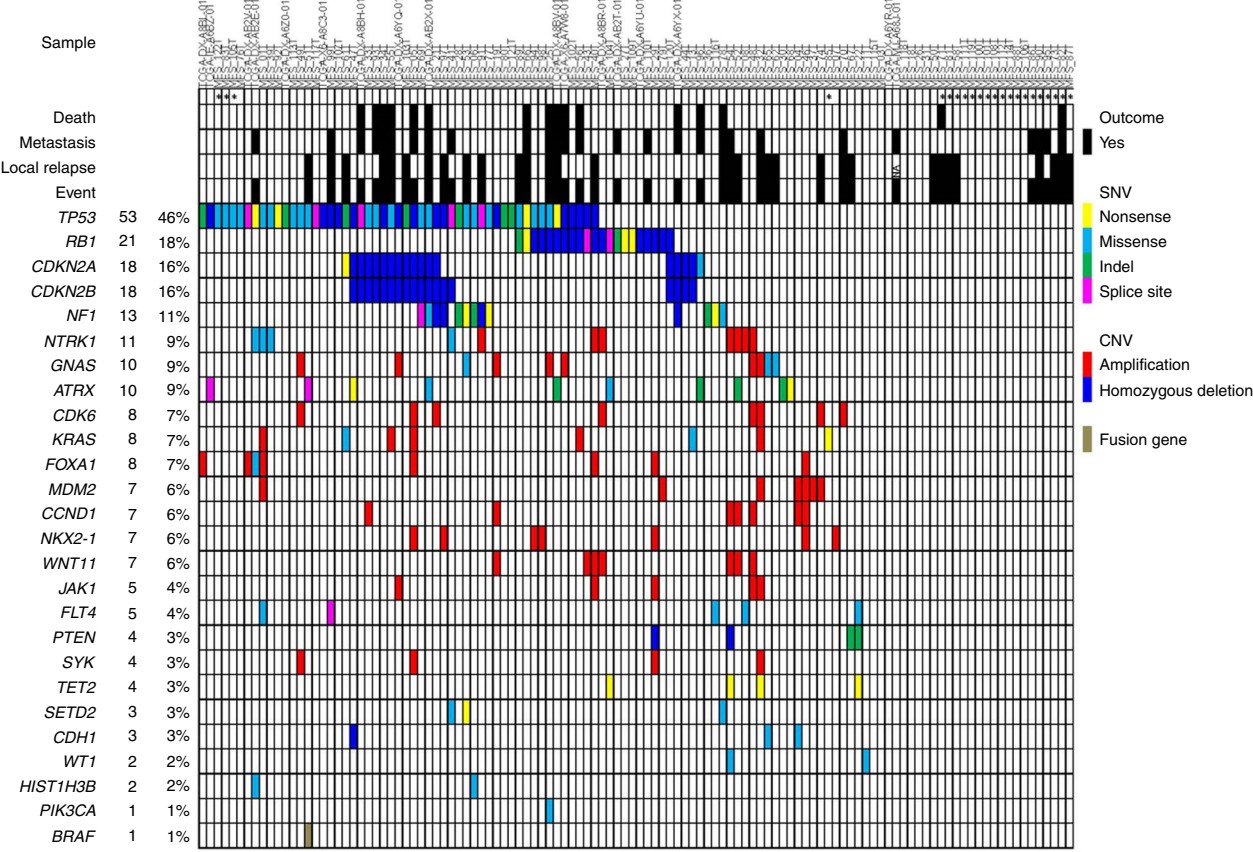

**Fig. 1** Driver gene landscape in 116 MFSs. Driver genes are ranked according to their frequencies of mutations. In each pathway, mutually exclusive profile was found: *RB1*, *CDKN2A/CDKN2B*, and *CCND1* in p53 signaling and the cell cycle G1/S checkpoint pathways; *NF1*, *NTRK1*, *KRAS*, *PTEN*, *SYK*, *PIK3CA*, and *BRAF* in the RTK-RAS-PI3K pathway; and *ATRX*, *TET2*, *SETD2*, *WT1*, and *HIST1H3B* in regulators of the epigenome

been previously reported in MFS, as well as alterations in known MFS driver genes, including *NTRK1*, *NF1*, and *PTEN*[7].

We also identified recurrent mutations in regulators of the epigenome, represented by *ATRX*, *TET2*, *SETD2*, *HIST1H3B*, and *WT1* (16%) (Fig. 1). ATRX, a member of the SWI/SNF family of chromatin remodeling proteins, is a histone chaperone and transcriptional regulator containing an ATPase/helicase domain[12]. It plays a role in depositing histones at heterochromatin and telomeric DNA[12–14]. Of *ATRX* mutations, eight (four frameshifts, two nonsenses, and two splice sites) of ten resulted in protein truncation (Fig. 2c) and disruption of the key ATPase domain, leading to predicted loss of function (Fig. 2d). ATRX cooperates functionally with DAXX, and *DAXX* mutations are also associated with ALT[13,15]. *DAXX* and *ATRX* mutations are mutually exclusive in PNET and uterine leiomyosarcoma[13,16,17]; however, no *DAXX* mutations were identified in our cohort.

In addition, all four *TET2* mutations were nonsense mutations resulting in truncated proteins lacking the functional double-stranded beta-helix (DSBH) domain (Fig. 2d).

**Comparison among soft tissue sarcomas with complex karyotype.** Although previous studies have partially revealed the importance of the p53 signaling pathway, cell cycle regulators, and the RTK-RAS-PI3K pathway in soft tissue sarcomas with complex karyotypes, such as MFS, dedifferentiated liposarcoma (DDLS), LMS, and UPS[7,18], a comprehensive overview of the molecular pathogenesis of these tumors, including differences among histological subtypes, has not previously been elucidated. Therefore, to construct a detailed map of the molecular alterations across diverse sarcoma subtypes, we analyzed data from an

additional 174 samples, including three major histologic subtypes of soft tissue sarcoma with complex karyotypes (50 DDLSs, 80 LMSs, and 44 UPSs), in addition to the 116 MFSs. The frequency of major genetic alterations in each subtype is provided in Supplementary Table 2. Although genes that belong to p53 signaling pathway, cell cycle regulators, and the RTK-RAS-PI3K pathway were widely affected in these sarcomas, each histological subtype exhibited a unique combination of mutation profile. In p53 signaling pathway, DDLS was distinctive in that *MDM2* was more frequently affected (94.0%, $p < 0.001$, $\chi^2$-test), whereas it was not the case for other three subtypes. In addition, high incidence of amplification of *CDK4* (86.0%, $p < 0.001$, $\chi^2$-test) made DDLS a more distinct entity compared to others in which *RB1* and *CDK6* were more frequently affected. In the RTK-RAS-PI3K pathway, *PTEN* was predominantly affected in LMS (18.8%, $p < 0.001$, $\chi^2$-test), whereas *NF1* and *KRAS* mutations were rare. By contrast, these two were predominantly seen in DDLS/MFS/UPS and DDLS/MFS, respectively. *NKX2-1* alterations were exclusively found in MFS ($p = 0.013$, $\chi^2$-test).

**Driver fusion gene in MFS.** In total, 29 MFS samples were subjected to RNA sequencing, resulting in identification of 1653 in-frame fusion transcripts (Supplementary Data 3). Among them, we identified a novel fusion gene involving the *BRAF* oncogene, *SLC37A3-BRAF*, in a single case (MFS_51; Fig. 3a). Expression of the *SLC37A3-BRAF* in-frame fusion transcript was confirmed by reverse transcription-PCR (RT-PCR) and Sanger sequencing. RNA sequencing demonstrated that the case carrying this fusion gene exhibited a higher expression of the *BRAF* transcript, with reads per kilobase of exon per million mapped

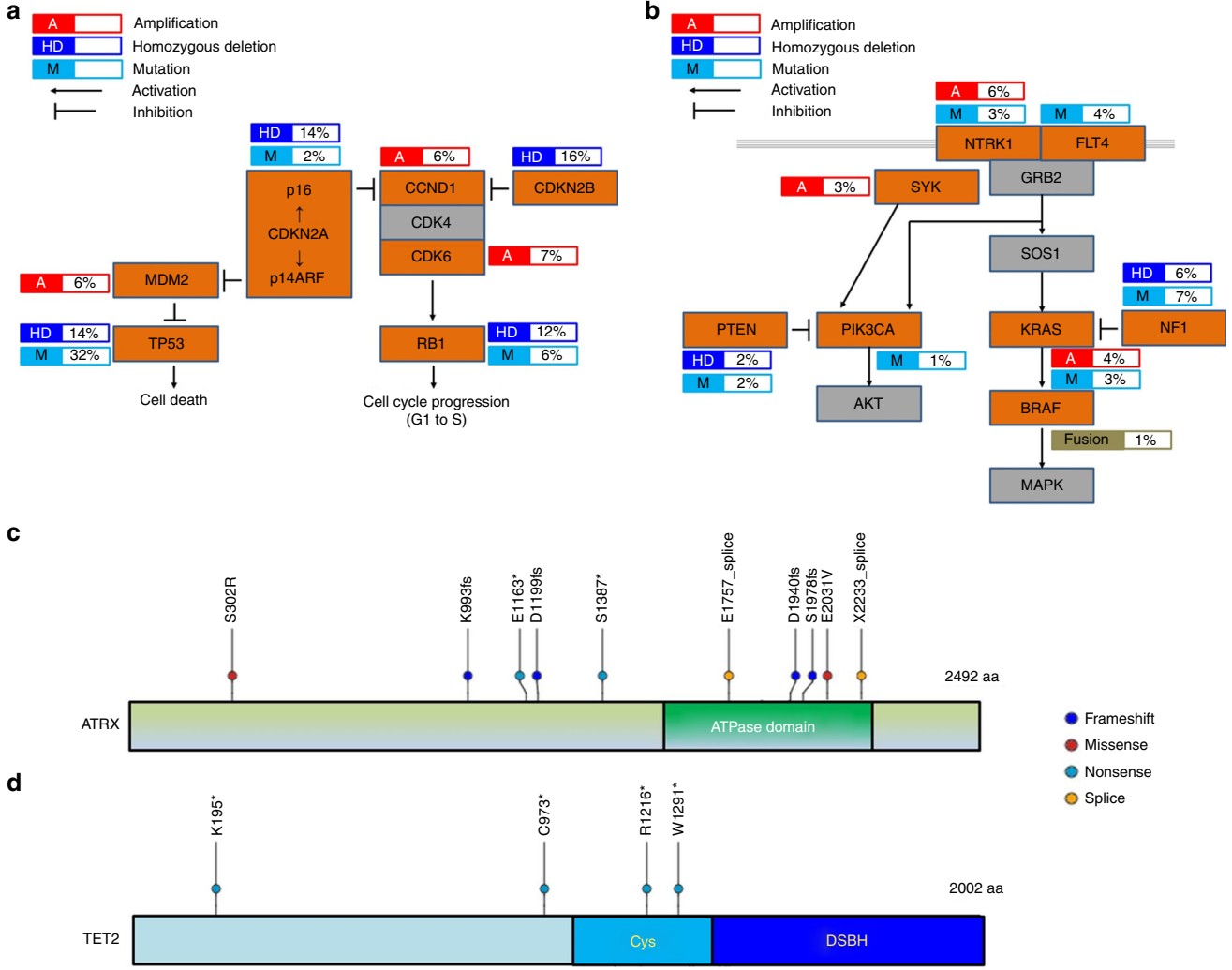

**Fig. 2** Recurrently altered pathways and genes in MFS. Frequently altered pathways in MFS: p53 signaling and cell cycle G1/S checkpoint pathways (**a**) and RTK-RAS-PI3K pathway (**b**). Genes with alterations are colored in orange. The types of alterations (amplification, homozygous deletion, or mutation) and their frequencies are indicated. Distributions of mutations of *ATRX* (**c**) and *TET2* (**d**) detected in 116 MFSs were shown. The location of somatic mutations in *ATRX* or *TET2* observed in this study is shown on the protein schematics. Types of mutations are featured by color as follows: frameshift indel, blue; missense, red; nonsense, light blue; and splice site, orange. The majority of mutations in *ATRX* were loss-of-function alterations that presumably lack the ATPase domain (**c**). All four nonsense mutations in *TET2* are predicted to encode truncated proteins lacking a functional DSBH domain (**d**). Cys cysteine-rich domain, DSBH double-stranded beta-helix domain

sequence reads value of 5.8 (interquartile range in all MFS cases, 2.9–5.4), suggesting that this fusion event induced increased expression of the *BRAF* gene. The majority of previously reported *BRAF* gene fusions retain the BRAF kinase domain, but lack the N-terminal RAS-binding domain, which is responsible for negative regulation of the BRAF kinase[19–21]; therefore, the *BRAF* gene fusion identified in this study is exceptional because it retains both the BRAF kinase and RAS-binding domains. Hence, we further investigated tumorigenic activity of this fusion transcript in vivo.

*SLC37A3-BRAF* cDNA ectopically expressed in mouse NIH3T3 fibroblast cells induced phosphorylation of ERK, a downstream mediator of BRAF (Fig. 3b). Subcutaneous transplantation of NIH3T3 cells expressing SLC37A3-BRAF, as well as those expressing the BRAF V600E mutation (a positive control), generated tumors in nude mice (Fig. 3c). Therefore, we conclude that the SLC37A3-BRAF fusion functions as a driver gene in MFS.

**Deregulated gene expression in MFS.** We investigated significantly upregulated and downregulated genes in 29 MFSs

(Supplementary Data 4a and 4b, respectively). The top-ranked genes included growth factors (*FGF8*, *BMP7*, *IGF2*, *TGFB2*, and *FGF18*), oncogene (*CTNNB1*), and immune-related molecules (*ICOS* and *TIGIT*) (Supplementary Table 4a and Supplementary Fig. 2). They could also be potentially novel targets for MFS. Although not in the top-ranked genes, previously reported prognostic genes including *CDK6*, *AMACR*, *SKP2*, *EZR*, and *MET*, were also significantly upregulated in our cohort[22–26].

**DNA methylation profiling and integrated analysis.** To further investigate molecular subtypes of MFS, genome-wide DNA methylation analysis was performed in 41 MFS tumor samples using an Infinium HumanMethylation450 BeadChip (Illumina). DNA methylation profiling based on unsupervised hierarchical clustering identified three MFS subtypes (Clusters A–C) with distinct methylation signatures (Fig. 4a).

Presence of unique genetic alterations was associated with these methylation subtypes (Fig. 4b). All *MDM2*-amplified MFS cases ($N = 5$) were in Cluster A ($p < 0.001$, $\chi^2$-test), whereas the

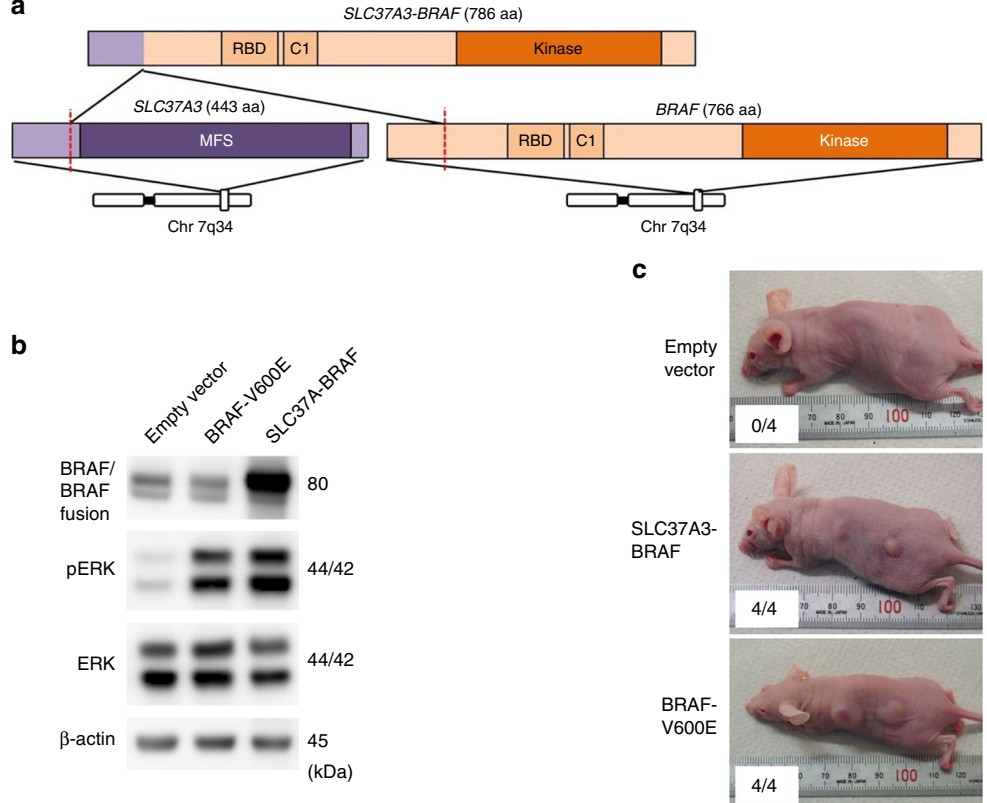

**Fig. 3** Oncogenic *BRAF* fusion gene detected in MFS. **a** Schematic presentation of the proteins encoded by *SLC37A3*, *BRAF*, and *SLC37A3-BRAF* fusion gene. MFS major facilitator superfamily, RBD RAS-binding domain, C1 phorbol esters/diacylglycerol-binding domain (C1 domain), Kinase protein kinase. **b** BRAF fusion induced ERK activation. Proteins extracted from cells transfected with retroviruses containing control, *BRAF-V600E*, and *SLC37A3-BRAF* were immunoblotted with antibodies against BRAF, phosphorylated ERK (p-ERK), ERK, and β-Actin. **c** Tumorigenicity of *SLC37-BRAF* fusion. Representative images of mice subcutaneously transplanted with NIH3T3 cells expressing empty vector or the *SLC37A3-BRAF* fusion. NIH3T3 cells transformed with BRAF V600E were used as a positive control

frequency of *TP53* alteration was significantly higher in Cluster B (42%) and C (82%) ($p = 0.010$, $\chi^2$-test). Homozygous deletions of *CDKN2A* were significantly more frequently found in Cluster C (46%) compared with Cluster A (9%) and B (11%) ($p = 0.040$, $\chi^2$-test). *NF1* alterations were significantly predominant in Cluster C (36%) compared with Cluster A (9%) and B (0%) ($p = 0.013$, $\chi^2$-test). All *ATRX* mutations were observed in Cluster B (21%) ($p = 0.077$, $\chi^2$-test), while Cluster A was characterized by alterations of genes associated with histone/chromatin modification (*TET2*, *SETD2*, and *WT1*) (46%) ($p = 0.013$, $\chi^2$-test). Especially, all MFS cases with *TET2* mutations were included in Cluster A ($p = 0.012$, $\chi^2$-test).

These clusters also correlated with clinical outcomes. Cases in Cluster C presented poorer overall survival, whereas no patient in Cluster A died as a result of MFS ($p = 0.040$, log-rank test) (Fig. 4c).

Next, we attempted to identify significantly hyper-methylated genes in each cluster, using a Wilcoxon rank-sum test ($q$ of false discovery rate <0.2) (Supplementary Data 5). Functional annotation analysis using the Database for Annotation, Visualization and Integrated Discovery (DAVID)[27] demonstrated that differentially methylated genes in Clusters B and C were enriched for homeobox genes compared with those in Cluster A (Supplementary Data 6a and b). In addition, genes encoding cadherins and fibronectins, which regulate cell adhesion and are important determinants of tumor progression, were markedly enriched in the differentially methylated genes of Clusters B and C (Supplementary Data 6a–c)[28–32]. Although these findings require validation in a larger number of independent samples, hyper-

methylation of these genes could potentially be used for discriminating MFS subtypes.

We further compared the methylation and expression data in 29 tumor samples with RNA sequencing. The methylation and expression data were generally negatively correlated, confirming little noise in our methylation analysis. Genes with significant negative correlation were summarized in Supplementary Data 7, and this analysis also revealed that promoter methylation silenced a wide range of genes including those associated with development, cell cycle and cell adhesion regulators, and homeobox genes.

We also investigated the relation between methylation clusters and upregulated genes (Supplementary Fig. 3). Cluster A was represented by upregulations of immune-related molecules and Cluster B was represented by upregulations of growth factors and *CTNNB1* (Supplementary Fig. 3). These correlations prompted us to infer the differences in immune cell fraction using CIBER-SORT[33] for independently and fully evaluating immune cell compositions among methylation-based clusters. There was significant difference of immune cell compositions among three clusters (Fig. 5). Especially, the average fraction of CD8+ T cell was significantly higher in better prognostic Cluster A ($p = 0.033$, $\chi^2$-test). There was no significant difference in the total numbers of non-silent mutations among the three clusters ($p = 0.804$, Student's $t$ test), and molecular mechanisms to recruit immune cells in specific cluster remain to be undetermined.

**Molecular prognostic factors in MFS.** No molecular prognostic factor has been so far identified in MFS; therefore, we investigated the impact of genetic alterations on patients' survival

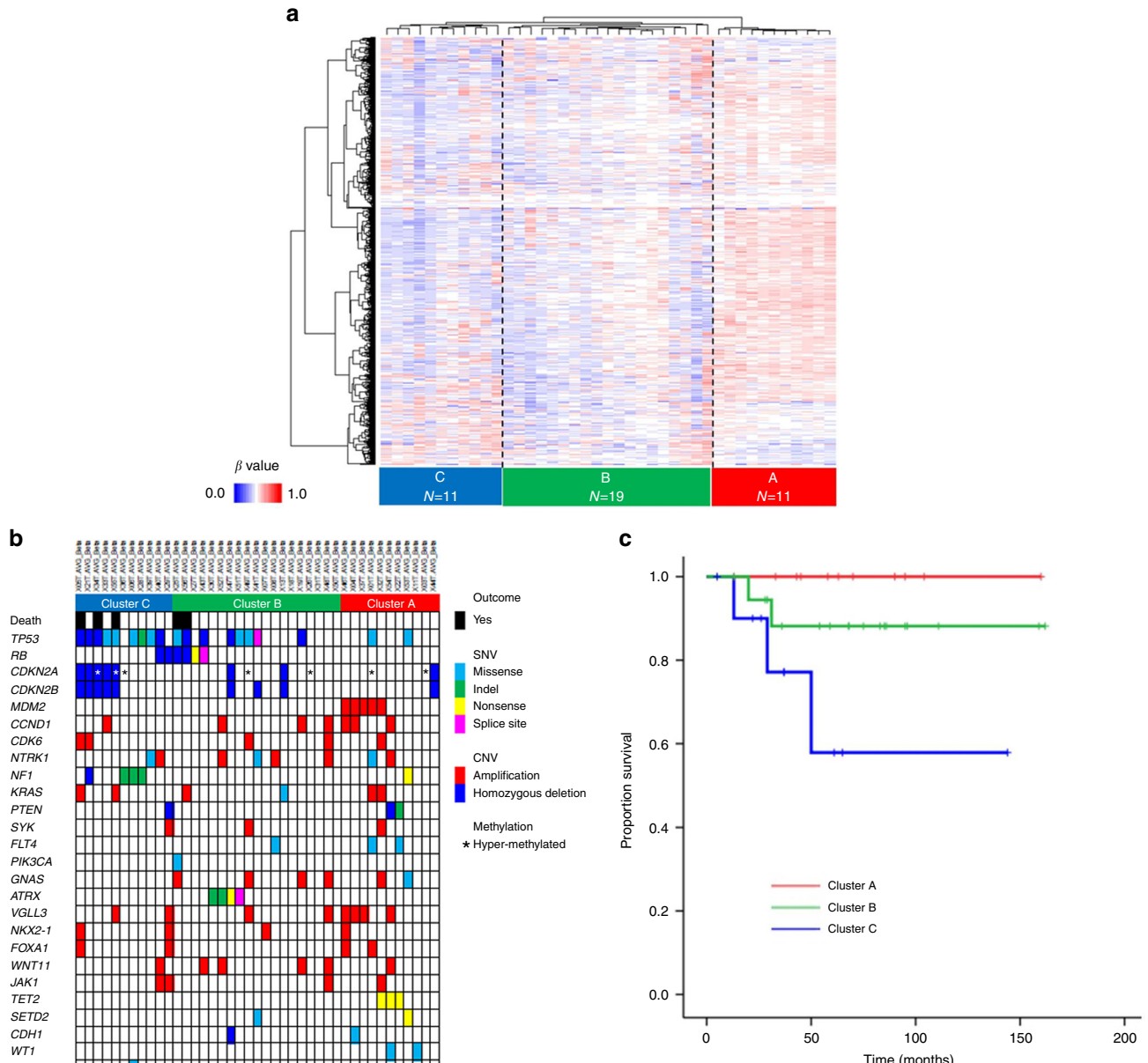

**Fig. 4** DNA methylation profiling and integrated analysis. **a** Unsupervised hierarchical clustering of DNA methylation in MFS. The heat map shows DNA methylation profiles of 41 MFSs. Three clusters were identified in 41 MFSs: Clusters A, B, and C. Red indicates hyper-methylation, and blue indicates hypo-methylation. **b** Correlations of DNA methylation clusters with genetic alterations. The horizontal axis represents each tumor. Each cluster was characterized by unique genetic alterations: *MDM2* amplification and alterations of histone/chromatin modifiers in Cluster A; *TP53* and *ATRX* alterations in Cluster B; and *TP53*, *CDKN2A/B*, and *NF1* alterations in Cluster C. **c** Kaplan–Meier plots of overall survival of patients segregated by three DNA methylation clusters. *p*-value was calculated by log-rank test

(Supplementary Figs. 4 and 5 and Supplementary Table 3). Alterations of any of cell cycle regulators (*RB1*, *CDKN2A*, *CDKN2B*, *CCND1*, and *CDK6*) were associated with poorer overall survival ($p = 0.001$, log-rank test) (Supplementary Fig. 4m). *RB1* ($p = 0.040$, log-rank test), *CDKN2A* ($p < 0.001$, log-rank test), and *CDKN2B* ($p < 0.001$, log-rank test) alterations were associated with poorer overall survival, whereas they were not with local recurrence-free survival. The presence of *TP53* alteration and *KRAS* amplification was also significantly associated with poorer overall survival ($p = 0.038$ and 0.010, respectively, log-rank test) (Supplementary Figs. 4 and 5). The presence of *GNAS* mutations was significantly associated with local recurrence-free survival ($p = 0.004$, log-rank test) (Supplementary Fig. 5l), whereas interestingly, none of *GNAS*-mutated cases

suffered from disease-associated death (Supplementary Fig. 4l). Of note, none of the *GNAS* mutations in MFSs involved the R201 locus that characterizes intramuscular myxoma.

Correlation between loss of functional mutation ATRX or DAXX and clinical outcome has been reported in various cancers[13,16,34–36]. Alternative lengthening of telomeres (ALT) resulting from ATRX/DAXX loss is associated with poor outcome in uterine leiomyosarcoma[34,35] and DDLS[36] among soft tissue sarcomas; however, no significant association between *ATRX* alteration and survival was observed in this MFS cohort.

## Discussion
Our comprehensive molecular analyses of 116 MFSs uncovered diverse genetic and epigenetic alterations and identified potential

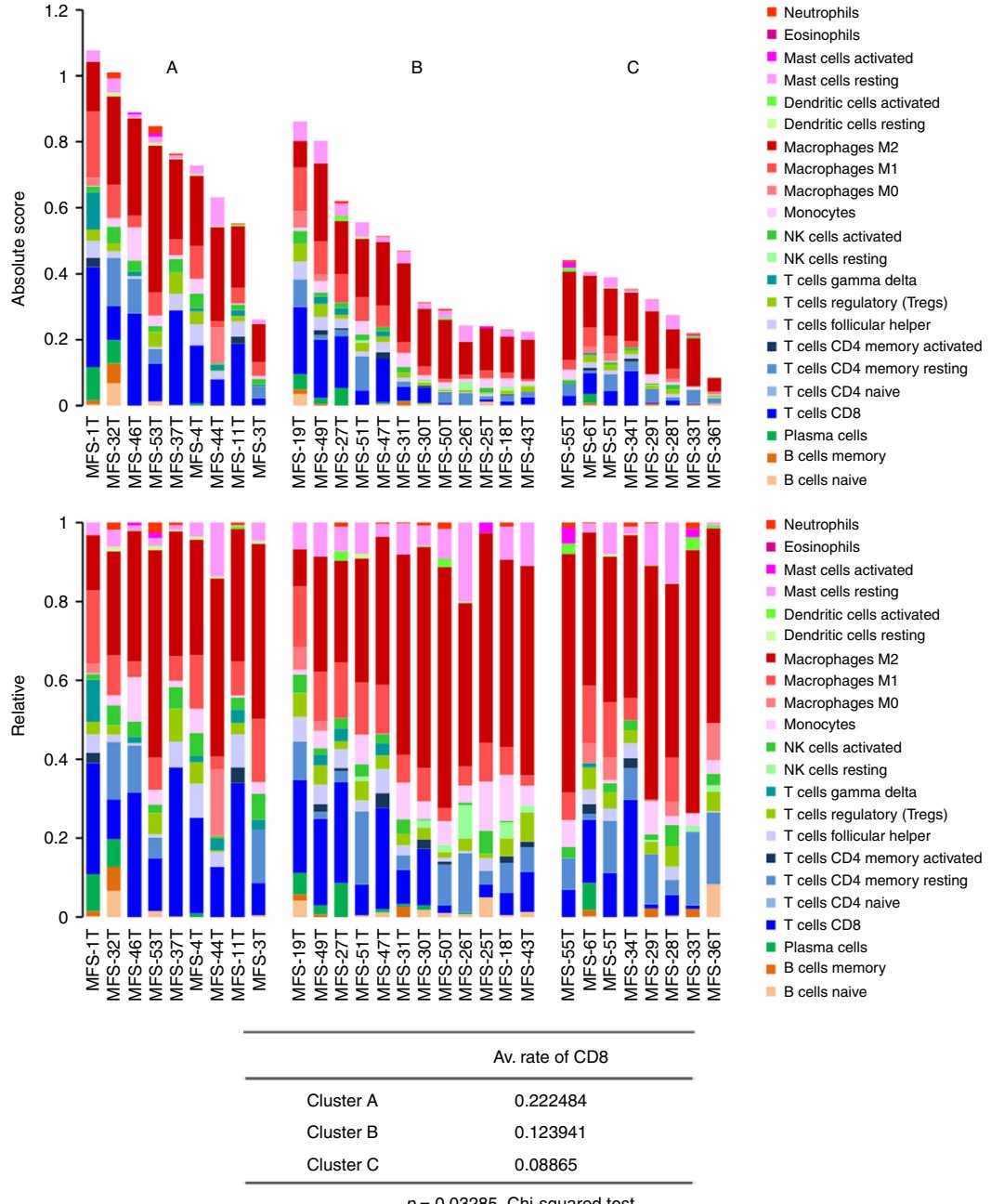

| | Av. rate of CD8 |
| --- | --- |
| Cluster A | 0.222484 |
| Cluster B | 0.123941 |
| Cluster C | 0.08865 |

*p* = 0.03285, Chi-squared test

**Fig. 5** Evaluation of immune cell fraction in MFS. Significant difference of immune cell composition was observed among three clusters with distinct methylation profiles. The average fraction of CD8+ T cell was significantly higher in Cluster A with better prognosis compared to others ($p = 0.033$, $\chi^2$-test)

therapeutic targets and molecular prognostic factors in this disease. Similar to previous reports of other soft tissue sarcomas[7,10,37], copy number alterations, as well as point mutations, appear to play important roles in MFS tumorigenesis. The present study identified novel recurrently mutated genes such as *GNAS* (9%), *ATRX* (9%), *KRAS* (7%), *CCND1* (6%), and *JAK1* (4%), in addition to previously reported mutated genes, including *TP53* (46%), *RB1* (18%), *CDKN2A* (16%), *CDKN2B* (16%), *NF1* (11%), *NTRK1* (9%), *MDM2* (6%), and *PTEN* (3%). We identified potential therapeutic targets in 39% of MFS cases, and further clinical evaluation of these targets, possibly by basket-type clinical trials, should be expected.

Mutations in *RB1*, *CDKN2A*/*CDKN2B*, and *CCND1* were almost completely mutually exclusive (Figs. 1 and 2a), and were

significantly correlated with clinical outcome. These data reveal a unique subset of MFS with aberrant cell cycling and poor prognosis for which intensification or novel therapeutic approaches should be considered. These biomarkers could be diagnostically useful because they can be detected by immunohistochemical evaluation of formalin-fixed, paraffin-embedded (FFPE) tissue samples. Our results also provide a rationale for the use of CDK4/6 inhibitors in MFS harboring genetic changes in the Rb pathway, including *CCND1* or *CDK6* amplification[38].

There were seven *MDM2*-amplified MFSs in our cohort. Although *MDM2* amplification is widely accepted as a marker for the diagnosis of DDLS in the internal trunk, even without associated well-differentiated liposarcoma (WDLS) component, it is still controversial whether *MDM2*-amplified undifferentiated

sarcoma at the peripheral site (including the truncal wall) should always be classified as DDLS in the absence of WDLS. Being aware that *MDM2* amplification is not restricted to DDLS[39], our approach in this study was to prioritize clinical and histological findings over *MDM2* status for tumor classification. All the seven *MDM2*-amplified tumors in our cohort developed at the peripheral sites (of which six arose in the superficial sites) and demonstrated classic histology of MFS with no evidence of associated WDLS component. Further, none of these tumors harbored *CDK4* co-amplification, unlike DDLS that is characterized by coexistence of *MDM2* (100%) and *CDK4* (92%) amplifications[10]. Moreover, no significant difference was observed with regard to the distribution of genetic alterations between *MDM2*-amplified and non-amplified MFS in our series, except for *TP53* alteration, which occurred almost mutually exclusively with *MDM2* amplification. Further studies are warranted to better understand the nosologic position of *MDM2*-amplified undifferentiated sarcoma at the peripheral sites.

*NF1*, a tumor-suppressor gene in which mutations cause the hereditary cancer predisposition disease, neurofibromatosis type 1, encodes a negative regulator of RAS proteins. Germline and somatic inactivation of *NF1* is associated with malignant peripheral nerve sheath tumors and gastrointestinal stromal tumors in individuals with neurofibromatosis type 1[40,41]. Seven somatic mutations in *NF1* were reported in soft tissue sarcomas; five in MFS and two in pleomorphic liposarcoma (PLS)[7]. We detected four homozygous deletions and nine somatic mutations of *NF1* in 116 MFSs. Seven of the nine somatic mutations of *NF1* caused premature truncation of the protein (Supplementary Fig. 6). These mutations located close to the previously reported mutations in MFS and PLS (Supplementary Fig. 6)[7]. These data suggest that a distinctive pattern of *NF1* aberrations may have a role in MFS tumorigenesis, similarly reported in other cancers[42,43]. Moreover, the use of MEK inhibitors may be a potential therapeutic option in NF1-deficient MFSs, as recent studies revealed that tumors harboring NF1 inactivation (inactivating/deleterious NF1 mutations) exhibited activation of the MAPK/ERK pathway and hence are potential targets for MEK inhibitors[44,45].

We identified other rare, but potentially actionable targets in MFS. Amplification of *JAK1* could be a therapeutic target in MFS because aberrant activation of the JAK/STAT pathway has been shown to be a promising target in various cancer types[46–48]. Other therapeutic targets included *NTRK1* and *NKX2-1*, both of which were discovered as targetable oncogenes and promising results in lung adenocarcinoma have been reported[49–53].

In addition, we identified a novel *BRAF* driver gene fusion, *SLC37A-BRAF*, which could be targeted with anti-BRAF therapies[19–21]. SLC37A3 is a solute carrier family 37, member 3 protein that has a transmembrane transporter activity and lacks putative protein-dimerization domain such as coiled-coil or zinc-finger domains. We speculated that overexpression of SLC37A-BRAF fusion protein depends on the promoter activity of *SLC37A3* because *SLC37A3* gene expression is specifically high in uterus, cervix, and fibroblast (GTEx Portal[54]) and mesenchymal stem cell (BioGPS[55]), whereas *BRAF* gene expression is low in fibroblast and mesenchymal stem cell. Although cell of origin of MFS remains to be unknown, this suggests that promoter swapping between *BRAF* and *SLC37A3* by structural rearrangement may increase expression of *SLC37A-BRAF* fusion gene compared to that of wild-type *BRAF*. Although *BRAF* fusion has been recently identified in a subset of myxoinflammatory fibroblastic sarcoma[56], our MFS with *BRAF* fusion showed no findings that may suggest relationship with this rare low-grade acral sarcoma.

Our RNA sequencing identified 71 in-frame recurrent fusion genes as well as many singleton ones in 29 MFSs. They included a

*TRIO* gene fusion that was reported previously[57]. In this report, RNA sequencing of 117 non-translocation-related sarcomas, including 15 MFSs, identified one recurrent *TRIO* fusion genes with various partners in seven cases including one MFS. *TRIO* fusion gene was also found in 1 of 17 MFSs in the TCGA data[10]. Although *TRIO* fusion genes were speculated to be associated with a transcriptomic program of immunity/inflammation, cell proliferation, and migration, their oncogenic roles have been still unclear. Further functional analysis of this recurrent gene fusion would permit to uncover therapeutic applications for MFS with *TRIO* rearrangements.

Efforts have been made in a range of cancers to elucidate the tumor–host immune interactions and provide prognostic predictors according to the recent advances in cancer immunotherapy[58–62]. Recent study analyzed immune cell infiltration in 203 soft tissue sarcomas, including 17 MFSs, based on the gene expression signatures and revealed diversities of immune microenvironments across sarcoma types. They reported higher infiltration score of macrophage in MFS[10] that was concordant with our results (Fig. 5). The immune microenvironment features were also reported to be associated with patients' survival in several sarcoma types[10]. In the current study, we found significantly different immune cell compositions among methylation clusters. Especially average fraction of infiltrated CD8+ T cell was significantly higher in the better prognostic cluster, suggesting prognostic significance of CD8+ T cell infiltration in MFS. Further investigation of the interactions between tumor and the host immune system in MFS would uncover prognostic biomarkers as well as targets of immunotherapy for MFS.

In conclusion, our comprehensive and integrative molecular analyses of MFS have uncovered substantial recurrent driver genes, most of which have not been identified as driver genes in MFS to date. DNA methylation patterns clustered into three subtypes closely associated with immune cell compositions, especially the fraction of CD8+ T cell, as well as unique combinations of driver mutations and prognosis. The driver alterations, significantly upregulated genes, immune cell compositions, and DNA methylation pattern identified in this study, would help to identify potential therapeutic targets as well as molecular subtypes with prognostic significance. These results are expected to provide a valuable basis for the development of precision medicine approaches, including molecular diagnostics and therapeutics, in MFS.

## Methods

**Patients and samples**. The study cohort comprised 116 cases with MFS: 99 cases from the current study and 17 from TCGA data set. Fresh-frozen tumor and normal tissues were provided by the NCC Biobank. Material for targeted sequencing was extracted from FFPE tissue blocks in selected cases. Macrodissection was performed on FFPE tumor tissues to ensure sample tumor content. Sequence data from the 17 MFSs from TCGA[10] were included after reviewing virtual hematoxylin and eosin-stained (HE) slides that are publicly available for review via cBioPortal website, and the diagnosis of 99 cases with MFS from the NCC Biobank was confirmed by pathological review (AY), according to the 2013 WHO classification. Specifically, we diagnosed MFS for pleomorphic spindle cell (or rarely epithelioid-appearing) sarcoma of uncertain/undifferentiated phenotype showing stromal myxoid change that accounted for at least 10% of the total tumor volume, according to WHO and Mentzel et al.[63]. Any evidence of unequivocal differentiation toward specific lines (e.g., lipoblasts) was considered incompatible with MFS. However, myofibroblastic differentiation represented by amphophilic fibrillary cytoplasm and the immunoexpression of actin and/or desmin (with no accompanying expression of myogenin or caldesmon) was accepted for the MFS designation. MFS mimics such as myxoinflammatory fibroblastic sarcomas and pleomorphic dermal sarcomas were excluded. Curvilinear vessels in the myxoid areas and diffuse infiltration along fascial planes were characteristic of MFS; however, such features were not absolute requirements for the diagnosis. *MDM2* amplification status was not considered in the review process, unless the tumor was located in the internal trunk. All tumor samples in the discovery set ($N = 41$) and TCGA data set ($N = 17$) were from primary tumors. Tumor samples in the validation set ($N = 58$) included 38 primary tumors, 17 recurrent tumors, and

3 metastatic specimens (Supplementary Fig. 7). The patient demographics and tumor characteristics of all cases ($N = 116$) are summarized in Supplementary Data 8.

DNA was extracted from fresh-frozen tissues or FFPE tissues using the standard phenol–chloroform method. DNA was quantified using a Qubit Fluorometer (Life Technologies, Carlsbad, CA, USA). The tissues and clinical information used in this study were obtained under informed consent and with the approval of the National Cancer Center Institutional Review Board.

**Whole-exome sequencing.** Whole-exome capture libraries were prepared from tumor and matched normal DNA, using a SureSelect Human All Exon V5+lncRNA kit (Agilent Technologies, Santa Clara, CA, USA) according to the manufacturer's instructions. DNA libraries were sequenced in a paired-end mode, using HiSeq 2000 platform (Illumina Inc., San Diego, CA, USA).

**Library preparation for exome sequencing.** Sequencing libraries were prepared, and captured using SureSelect Human All Exon V5+lncRNA kit (Agilent Technologies, Santa Clara, CA, USA) following the manufacturer's instructions. In brief, 1 µg of DNA was fragmented using a Covaris S2 system (Covaris Inc. Woburn, MA, USA) to produce fragments with an average size of 150–200 base pairs, followed by end repair, A-tailing, and ligation of SureSelect adapter oligos using SureSelect XT Reagents Kit (Agilent Technologies). Precapture polymerase chain reaction (PCR) amplification of the adapter-ligated library was performed for 5–6 cycles. Next, 750 ng of the amplified libraries were hybridized with the SureSelect capture library for 24 h and purified. Post-capture amplification was performed for 10–11 cycles using Herculase II polymerase with SureSelect Indexing Post-Capture PCR primers. Purification was performed with Agencourt AMPure XP beads (Beckman Coulter, Brea, CA, USA). Quantification and size distribution of the amplified libraries were determined using a BioAnalyzer (Agilent Technologies) and KAPA Library Quantification Kit (KAPA Biosystems).

**Mutation calling.** Paired-end reads were aligned to the human reference genome (GRCh37) using the Burrows–Wheeler Aligner (BWA)[64] for both tumor and non-tumor samples. Probable PCR duplications, in which paired-end reads aligned to the same genomic positions, were removed, and pile-up files were generated using SAMtools[65] and a program developed in house. As sequence errors occur in a sequence-specific manner, the read information from all non-tumor samples was pooled into a so-called normal panel to accurately discriminate true positives from false positives. The details of our filtering conditions were reported previously[66].

**Processing significantly mutated genes (SMGs).** SMGs were estimated by aggregating somatic substitutions, short indels, focal amplifications, and homozygous deletions (aggregated mutation method, AMM). The expected number of each type of mutation in each gene was estimated from the background mutation rate, and tests of significance for each gene were performed by assuming a Poisson distribution ($p$ value). Multiple testing adjustments ($q$ value) were performed using the method described by Benjamini and Hochberg[67]. In this study, SMGs were accumulated using three separate tests, as reported previously[66] (AMM, activation bias method, and inactivation bias method), and the results were combined in a final table of SMGs (Supplementary Tables 1 and 4). The $q$ values from each individual significance test were combined as the geometric mean. A test was also developed to detect bias in the mutation allele fraction (subclonal bias), which, if consistently low, would suggest a gene was a passenger rather than a driver. Genes in the final combined table that failed this bias test were removed from the final list of SMGs.

Initial copy number (CN) was estimated by comparing read depth of tumor and normal samples. The initial CN estimates were adjusted by tumor purity and tumor ploidy of each sample and calculated using an in-house program[68]: adjusted CN = $[1 - (1 - R(x))/\text{tumor purity}]*\text{tumor ploidy}$, where $R(x)$ is the ratio of tumor read depth to normal read depth at probe $x$. Then GISTIC version 2.0[69] was used to calculate significant CN gains and losses. $\log_2(\text{adjusted CN})$ values for all probes were calculated and segmented using DNACopy[70]. GISTIC was performed using the segmented CN. First, we selected genes with GISTIC $q$ value <0.25 or associated with MFS reported in previous literature[7] or TCGA data set for significantly mutated gene analysis. Focal amplifications (AMPs) were selected if adjusted CN was ≥6. Homozygous deletions (HDs) were selected if adjusted CN was equal to or less than 0. When only few somatic mutations and heterozygous SNPs were observed, it is difficult to estimate accurately the tumor purity and adjusted CN may not be accurate. Therefore, we removed false positive HDs with CN before adjustment greater than 0.5. This does not represent heterozygous in this filtering. It represents the CN before adjustment when tumor purity is 0.5 (average tumor purity in our cohort) and adjusted CN is 0.

**Targeted sequencing.** Following DNA extraction and quantification, we measured the amplification of *RNaseP*, a housekeeping gene, in each FFPE-derived sample and calculated the "amplified *RNaseP* concentration/Qubit-based DNA concentration ratio (*RNaseP*/Qubit ratio)" for each sample (manuscript in preparation). We used this ratio as an index of DNA amplifiability, which presumably correlates inversely with the extent of FFPE-derived DNA degradation. In fact, a

correlation was observed between this index and the peak size (bp) of the Pre-Capture PCR product (data not shown). We sheared DNA (input: 200–600 ng) via sonication using a Covaris S2 system (Covaris Inc.). During library preparation, we observed that library size was small when the *RNaseP*/Qubit ratio was low (data not shown). After noticing this trend, we skipped the shearing process for such samples ($n = 16$ non-sheared samples).

Sequenced libraries were prepared using a combination of the Hyper Prep Kit (Kapa Biosystems, Wilmington, MA, USA) and SureSelect XT Library Prep Kit (Agilent Technologies); the library enrichment procedures suggested by manufacturers were generally followed, albeit with two modifications: (i) adjusted the SureSelect Adapter Oligo Mix volume according to the DNA input volume, and (ii) we used twice the recommended primer volume to ensure better PCR amplification. PreCapture PCR products (just before capture hybridization) with an approximate peak size of 200–250 bp (range, 162–313 bp) were quantified using a BioAnalyzer DNA1000 chip (Agilent Technologies). The capture regions were designed to cover coding exons with an additional 10 bases on the 3′ and 5′ ends. The total targeted DNA length was 418 kbp. A custom SureSelect bait library (Agilent Technologies) was used to capture all exons of 140 candidate genes, selected according to the data from exome sequencing, including recurrently mutated genes identified by exome sequencing and their associated genes, and a review of the published literature[7,11] (Supplementary Table 1). Prepared libraries were sequenced on Illumina HiSeq 2000 ($N = 195$) and MiSeq ($N = 3$) platforms. Paired-end reads were trimmed to remove adapter contamination using the Cutadapt software package prior to alignment with the human reference genome (GRCh37). After variant calling, mutations with a strand bias (between forward and reverse reads) >95% were removed.

For target sequencing data, CN of each gene was estimated based on the relative read depth of the coding regions of paired normal and tumor samples (CN ratio, thereafter). The CN of each gene was normalized by dividing the average read depth of the coding region by the average read depth of all coding regions. Tumor purity was estimated from the average allele frequency of somatic mutations. Focal amplifications and homozygous deletions were selected using the same conditions in WES data.

**RNA sequencing.** RNA sequencing was performed using 29 tumor samples with high-quality RNA (RNA integrity number >6.5) on a HiSeq 2000 instrument. RNA-sequencing libraries were prepared using a SureSelect Strand-Specific RNA Library Prep Kit (Agilent Technologies), following the manufacturer's instructions. Paired-end 100 bp reads were mapped and aligned to known RNA sequences in the RefSeq, Ensembl, and LincRNA databases using the BWA-MEM program. Candidate fusion transcripts were identified using an in-house algorithm[68].

The Bowtie program was performed with the—v3 option so that three or fewer mismatches were allowed and with the—a option so that all multiple hits could be detected, as there are many splice variants in RNA databases. After selecting the best hits with the proper spacing and orientation, RPKM values were calculated.

Paired-end reads of 50 bp in length were used for fusion gene detection because they contain longer spacers than 100-bp paired-end reads. The 100-bp paired-end reads were cut to generate 50-bp paired-end reads. To detect fusion genes, paired-end reads aligned uniquely to the different genes with two mismatches or fewer were considered, and paired clusters indicating fusion transcripts were selected as follows. Forward and reverse clusters, which included paired-end reads, were constructed from the end sequences aligned in each respective direction. Two reads were allocated to be in the same cluster if their end positions were not farther apart than 300 bp.

Clusters with greater than 1000 bp between the left-most and right-most reads were discarded. Paired-end reads were selected if one end sequence fell within the forward cluster and the other end sequence fell within the reverse cluster (hereafter, these pairs of forward and reverse clusters are referred to as "paired clusters"). Paired clusters that included at least one pair of paired-end reads perfectly matched to human reference RNA sequences were selected, and gene pairs mis-selected owing to nucleotide variations were removed. For this purpose, paired-end reads included in paired clusters were aligned to human reference RNA sequences using the BLASTN program. If one end sequence was aligned to a region of paired clusters and the other end was aligned to the same RNA sequences with the proper spacing and orientation of the paired-end library, the gene pair was removed. An expectation value of 1000 was used as a cutoff value for BLASTN so that paired-end reads with low similarity to the human reference RNA sequences could also be aligned. Finally, gene pairs with two paired-end reads or more for in-frame fusion genes and three paired-end reads or more for fusion genes other than in-frame were selected.

**RT-PCR and Sanger sequencing for *BRAF* gene fusion.** The status of the candidate *SLC37A3-BRAF* fusion gene in sample MFS_51T was examined by RT-PCR followed by Sanger sequencing. Total RNA (200 ng) of MFS-51T was reverse-transcribed into cDNA using iScript cDNA synthesis kit (BioRad). cDNA was subjected to PCR amplification using TaKaRa Ex Taq Hot Start Version (TaKaRa). The reactions were carried out in a thermal cycler under the following conditions: 95 °C for 2 min, 40 cycles of 98 °C for 5 s, 60 °C for 40 s, and 72 °C for 40 s, with a final extension at 72 °C for 5 min. The gene encoding glyceraldehyde-3-phosphate dehydrogenase (*GAPDH*) was amplified to estimate the efficiency of cDNA

synthesis. The amplified products were sequenced using the BigDye Terminator v3.1 Cycle Sequencing Kit (Life Technologies) and analyzed on a 3130xl Genetic Analyzer (Life Technologies). The PCR primers used in this study are as follows: F (5′-CTGCTCACTTTCTTCAGTTATTCGTT-3′) and R (5′-CTAGCTTGCTGGTGTATTCTTCATAG-3′).

Gene expression was quantified as described previously[68]. After removing the gene mapped no read in over 50% of samples, and converted RPKM value to $log_2$, we applied Welch's $t$-test to every genes by comparing the two groups; six higher/lower-expressing samples (20% of the cases) and others (80% of the cases) for identifying up/downregulated genes. The correlation between methylation and expression level was tested by Spearman's correlation test. The RPKM data were used for Immune cell fraction analysis using CIBERSORT[33] with LM22 as a signature file.

**DNA methylation analysis**. Comprehensive DNA methylation analysis was performed for 41 tumor samples undergoing WES using the Infinium Human-Methylation450 BeadChip (Illumina), which covered 482,421 CpG and 3091 non-CpG sites, according to the manufacturer's protocol. The methylation level of CpG sites was represented by $\beta$ values ranging from 0 (completely unmethylated) to 1 (completely methylated). $\beta$ values were normalized to generate $M$ values[71], which were used for subsequent analyses.

For unsupervised hierarchical clustering analysis, 1918 probes with absolute $M$ values of the coefficient of variance >60 were first selected. Then, hierarchical clustering using the heatmap.3 function in R package was performed[72], with the Euclidean distance as the metric and Ward's method as the agglomeration method.

**Functional evaluation of the *SLC37A3-BRAF* fusion gene**. The full-length *SLC37A3-BRAF* fusion and *BRAF* V600E mutant cDNA sequences were chemically synthesized (Eurofins Genomics) and subcloned into pMXs-neo vector (Cell Bio-labs) to generate recombinant retroviruses. Mouse NIH3T3 fibroblast cells, obtained from the American Type Culture Collection (ATCC) and tested for mycoplasma, were infected, and clones with stable expression were established. Whole-cell lysates were subjected to immunoblot analysis using primary antibodies against BRAF (1:500; #14814, Cell Signaling Technology), p44/42 ERK (1:500; #4695, Cell Signaling Technology), and phosphorylated p44/42 ERK (Thr202/Tyr204) (1:500; #4370, Cell Signaling Technology). Original images of the immunoblots were shown in Supplementary Fig. 8. To analyze tumorigenic activity of the BRAF fusion gene and BRAF mutant, a total of $1 \times 10^6$ transduced NIH3T3 cells were injected subcutaneously into nude mice (BALB/c-nu/nu, CLEA Japan). Tumor formation was measured after 21 days. All mouse procedures were performed with the approval of the Animal Ethics Committee of the National Cancer Center, Tokyo, Japan.

**Statistical analysis for survival analysis**. All time-to-event end points were calculated using the Kaplan–Meier method. Survival analysis was calculated from the day of diagnosis, and potential prognostic factors were identified by univariate analysis using the log-rank test. Prognostic factors were evaluated using the Cox proportional hazards regression model.

The number of non-silent mutations among methylation clusters were compared using the Student $t$ test. Distributions of gene alterations among the methylation clusters were compared using $\chi^2$-tests. The threshold for statistical significance was $p < 0.05$. All statistical analyses were conducted using IBM SPSS version 19.0 (IBM SPSS, Armonk, NY, USA).

**Data availability**. Accession codes. Whole-exome sequencing data and RNA sequencing data were deposited in the European Genome-phenome Archive (EGA) database under accession EGAS00001002889. The methylation microarray data were deposited in the Gene Expression Omnibus (GEO) database under accession GSE111200. All other relevant data are available from the authors.

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

# ARTICLE

36. Lee, J. C. et al. Alternative lengthening of telomeres and loss of ATRX are frequent events in pleomorphic and dedifferentiated liposarcomas. *Mod. Pathol.* **28**, 1064–1073 (2015).

37. Kanojia, D. et al. Genomic landscape of liposarcoma. *Oncotarget* **6**, 42429–42444 (2015).

38. Finn, R. S. et al. The cyclin-dependent kinase 4/6 inhibitor palbociclib in combination with letrozole versus letrozole alone as first-line treatment of oestrogen receptor-positive, HER2-negative, advanced breast cancer (PALOMA-1/TRIO-18): a randomised phase 2 study. *Lancet Oncol.* **16**, 25–35 (2015).

39. Wallander, M. L., Tripp, S. & Layfield, L. J. MDM2 amplification in malignant peripheral nerve sheath tumors correlates with p53 protein expression. *Arch. Pathol. Lab. Med.* **136**, 95–99 (2012).

40. King, A. A., Debaun, M. R., Riccardi, V. M. & Gutmann, D. H. Malignant peripheral nerve sheath tumors in neurofibromatosis 1. *Am. J. Med. Genet.* **93**, 388–392 (2000).

41. Maertens, O. et al. Molecular pathogenesis of multiple gastrointestinal stromal tumors in NF1 patients. *Hum. Mol. Genet.* **15**, 1015–1023 (2006).

42. Ding, L. et al. Somatic mutations affect key pathways in lung adenocarcinoma. *Nature* **455**, 1069–1075 (2008).

43. Cancer Genome Atlas Research Network. Comprehensive genomic characterization defines human glioblastoma genes and core pathways. *Nature* **455**, 1061–1068 (2008).

44. Cancer Genome Atlas Research Network. Comprehensive molecular profiling of lung adenocarcinoma. *Nature* **511**, 543–550 (2014).

45. Arcila, M. E. et al. MAP2K1 (MEK1) mutations define a distinct subset of lung adenocarcinoma associated with smoking. *Clin. Cancer Res.* **21**, 1935–1943 (2015).

46. Xiang, Z. et al. Identification of somatic JAK1 mutations in patients with acute myeloid leukemia. *Blood* **111**, 4809–4812 (2008).

47. Mullighan, C. G. & Downing, J. R. Genome-wide profiling of genetic alterations in acute lymphoblastic leukemia: recent insights and future directions. *Leukemia* **23**, 1209–1218 (2009).

48. Balko, J. M. et al. Triple-negative breast cancers with amplification of JAK2 at the 9p24 locus demonstrate JAK2-specific dependence. *Sci. Transl. Med.* **8**, 334ra353 (2016).

49. Saito, M. et al. Gene aberrations for precision medicine against lung adenocarcinoma. *Cancer Sci.* **107**, 713–720 (2016).

50. Vaishnavi, A. et al. Oncogenic and drug-sensitive NTRK1 rearrangements in lung cancer. *Nat. Med.* **19**, 1469–1472 (2013).

51. Farago, A. F. et al. Durable clinical response to entrectinib in NTRK1-rearranged non-small cell lung cancer. *J. Thorac. Oncol.* **10**, 1670–1674 (2015).

52. Yamaguchi, T. et al. ROR1 sustains caveolae and survival signalling as a scaffold of cavin-1 and caveolin-1. *Nat. Commun.* **7**, 10060 (2016).

53. Yamaguchi, T. et al. NKX2-1/TITF1/TTF-1-induced ROR1 is required to sustain EGFR survival signaling in lung adenocarcinoma. *Cancer Cell* **21**, 348–361 (2012).

54. Battle, A., Brown, C. D., Engelhardt, B. E. & Montgomery, S. B. Genetic effects on gene expression across human tissues. *Nature* **550**, 204–213 (2017).

55. Wu, C. et al. BioGPS: an extensible and customizable portal for querying and organizing gene annotation resources. *Genome Biol.* **10**, R130 (2009).

56. Kao, Y. C. et al. Recurrent BRAF gene rearrangements in myxoinflammatory fibroblastic sarcomas, but not hemosiderotic fibrolipomatous tumors. *Am. J. Surg. Pathol.* **41**, 1456–1465 (2017).

57. Delespaul, L. et al. Recurrent TRIO fusion in nontranslocation-related sarcomas. *Clin. Cancer Res.* **23**, 857–867 (2017).

58. Bindea, G. et al. Spatiotemporal dynamics of intratumoral immune cells reveal the immune landscape in human cancer. *Immunity* **39**, 782–795 (2013).

59. Rooney, M. S., Shukla, S. A., Wu, C. J., Getz, G. & Hacohen, N. Molecular and genetic properties of tumors associated with local immune cytolytic activity. *Cell* **160**, 48–61 (2015).

60. Angelova, M. et al. Characterization of the immunophenotypes and antigenomes of colorectal cancers reveals distinct tumor escape mechanisms and novel targets for immunotherapy. *Genome Biol.* **16**, 64 (2015).

61. Galon, J. et al. Cancer classification using the Immunoscore: a worldwide task force. *J. Transl. Med.* **10**, 205 (2012).

62. Li, B. et al. Comprehensive analyses of tumor immunity: implications for cancer immunotherapy. *Genome Biol.* **17**, 174 (2016).

63. Mentzel, T. et al. Myxofibrosarcoma. Clinicopathologic analysis of 75 cases with emphasis on the low-grade variant. *Am. J. Surg. Pathol.* **20**, 391–405 (1996).

64. Li, H. & Durbin, R. Fast and accurate short read alignment with Burrows-Wheeler transform. *Bioinformatics* **25**, 1754–1760 (2009).

65. Li, H. et al. The sequence Alignment/Map format and SAMtools. *Bioinformatics* **25**, 2078–2079 (2009).

66. Totoki, Y. et al. Trans-ancestry mutational landscape of hepatocellular carcinoma genomes. *Nat. Genet.* **46**, 1267–1273 (2014).

67. Benjamini, Y. & Hochberg, Y. Controlling the false discovery rate: a practical and powerful approach to multiple testing. *J. R. Stat. Soc. Ser. B Stat. Methodol.* **57**, 289–300 (1995).

68. Nakamura, H. et al. Genomic spectra of biliary tract cancer. *Nat. Genet.* **47**, 1003–1010 (2015).

69. Mermel, C. H. et al. GISTIC2.0 facilitates sensitive and confident localization of the targets of focal somatic copy-number alteration in human cancers. *Genome Biol.* **12**, R41 (2011).

70. Olshen, A. B., Venkatraman, E. S., Lucito, R. & Wigler, M. Circular binary segmentation for the analysis of array-based DNA copy number data. *Biostatistics* **5**, 557–572 (2004).

71. Du, P. et al. Comparison of beta-value and M-value methods for quantifying methylation levels by microarray analysis. *BMC Bioinformatics* **11**, 587 (2010).

72. Gentleman, R. C. et al. Bioconductor: open software development for computational biology and bioinformatics. *Genome Biol.* **5**, R80 (2004).

## Acknowledgements

This study was funded by a Grant-in-Aid for Scientific Research from the Ministry of Education and Science (B, no. 22390296), and by the National Cancer Center Research and Development Fund (28-A-16 and 29-A-6). The National Cancer Center Biobank is supported by the National Cancer Center Research and Development Fund, Japan.

## Author contributions

Study design: K.O., A.K., A.Y., Y.T., and T.S. Sequence data production: K.O., F.H., and E.A. Data analysis: H.N., Y.T., N.H., M.N., and M.K. Statistical analysis: H.N., Y.T., N.H., M.N., and M.K. Molecular analysis: K.O., F.H., W.M., H.R., and Y.A. Sample acquisition and clinical data collection: K.O., A.Y., and A.K. Manuscript writing: K.O., H.N., Y.T., N. H., M.N., M.K., F.H., A.Y., and T.S. Project oversight: F.H., Y.A., Y.T., S.T., and T.S.

## Additional information

**Competing interests:** The authors declare no competing interests.

