## [Peer Review File · Nature Communications]

Reviewer #1 Expert in myxofibrosarcoma:

This study describes the epigenetic and genetic characterization of a large cohort of human myxofibrosarcoma tumor samples. The study combines whole exome sequencing and targeted sequencing to identify the most frequent mutations in MFS, and also performs RNA-seq and methylation analysis in a subset of these samples. The authors utilize these data to identify recurrent somatic mutations, copy number variation, novel fusion products, methylation clustering and potential prognostic factors in MFS. Interestingly, the authors identify a novel BRAF gene fusion in one sample and perform experiments to validate it as a driver mutation in MFS.

Major Points.

1. The authors should provide a flowchart to clarify the order in which different types of sequencing occurred and how many samples received each type of sequencing.
2. The authors describe generation of whole transcriptome data for 29 MFS samples, but provide analysis of only fusions gene identified. It would be interesting to see an analysis of any genes that were significantly upregulated or downregulated.

Minor Points:

1. line 37: The authors should reword the sentence, as it appears to suggest that 99 MFS samples underwent WES, but this is not the case based on later descriptions of sequencing.
2. In supplementary table 1, the authors should clarify the way in which the q-value is produced. It is unusual that so many of the q values are equal to one, and it unclear how this q-value relates to the p-value.
3. In line 172 the authors should reword sentence, as TP53 is the most frequently gene encoding a protein involved in cell cycle regulation in their MFS sample.
4. Minor spelling change: change to exclusive
5. The authors should clarify the selection criteria used for choosing the MFS samples that would undergo RNA-sequencing and/or methylation array analysis.
6. The authors should provide information how the quantity of transcript for the SLC37A3-BRAF found in the MFS samples compares to V600E BRAF samples, as the western blot appears to show significantly more SLC37A3-BRAF protein when compared to the V600E lane.
7. An effort should be made to identify any samples that were part of both methylation array and RNA seq experiments. Identifying highly methylated genes that have correspondingly reduced transcription levels would help validate methylation results and remove some noise from the methylation analysis.

Reviewer #2 Expert in myxofibrosarcoma genomics:

While this manuscript is well written, and provides a wealth of potentially interesting findings, particularly in the supplemental tables, I found the text itself to focus perhaps a little too much on alterations in the usual suspects (TP53, RB, RAS, AKT pathways), and not on highlighting more novel or "outside the box" types of observations, with some exceptions to this being ATRX and TET2. As an example, while table S7 highlights many potential in-frame gene fusions, the text drills down to focus on just one, involving BRAF, a known oncogene, seen in one case. While it is undoubtedly interesting that this oncogenic fusion exists, I would have been more excited to see a mechanistic study of some of the fusions that were present in multiple cases rather than the sole BRAF fusion, or even just speculation as to whether they might be functional or merely bystander fusions (e.g. the 10 cases with B2microglobulin fusions or the 13 cases with GOSR1 fusions, among other possibilities). In addition, there are problems with the methodology which are detailed in more depth below.

Major concerns.

1. The biggest issue I have with this study is the uncertainty I have over the actual diagnosis of the cases included in this study as MFS. The diagnosis of MFS can be challenging and

controversial, and it is unclear which criteria were used to diagnose the tumors for this study. While the WHO was cited, it would be helpful to have some specific criteria outlined in the methods. I am deeply dubious about the diagnosis of the cases which later are reported to have MDM2 amplifications, and suspect these to be dedifferentiated liposarcoma (DDLPS). These should be excluded from the study or included as reclassified DDLPS in table S6. Please describe the pathology review that was performed for case inclusion and diagnostic inclusion/exclusion criteria for these cases – simply citing the WHO is insufficient here. Also, it would be important to note whether these tumors are primary, recurrent, metastatic disease, or if the specimens were previously subjected to radiation or chemotherapy as neoadjuvant therapy and disease status may affect the findings. I note that the text does specify that WES and methylation studies included only primary, but this should be stated for all cases. (The only place the tumor status appears is in supplemental table 13 – did you see any difference in mutations between primary and metastatic/recurrent tumors).

2. The 25 “MFS” cases the authors have used from the TCGA dataset have not been subjected to any sort of QC protocol for this study. I am aware that the curated TCGA dataset will not be available until after that manuscript is published, nevertheless, nearly half of these 25 cases have poor quality data (high level of artifacts) or should not be considered MFS after adequate pathology review.

3. I am concerned about the methodology used to determine amplification/deletion in tumors with targeted sequence data only (which represent the majority of cases in the study), as well as the definition of “homozygous deletion” for both WES and targeted sequence data. In the methods, it states that tumors were assumed to be diploid when determining copy number. In my experience, this is not a safe assumption to make in MFS or any other high grade complex karyotype sarcoma. I would strongly urge the authors to re-evaluate their methodology here. Also, the definition of homozygous deletion is imprecise. Line 378 “ratio of tumor read depth to normal read depth was <0.50 ” – this could also be low purity even with a low level deletion. I would accept that this methodology would catch heterozygous deletions, but am not convinced it is sufficient to determine these to be homozygous. Line 380 “deletion affecting a known driver gene”. There is no reason to presume that a deletion affecting a known driver gene has to be homozygous. Tumor suppressors are frequently bi-allelically mutated and deleted. The use of circular terminology in defining deletions here minimizes the fact that heterozygous deletions are at least as common as homozygous in these tumors, yet nowhere are heterozygous deletions mentioned or displayed in this work.

4. Overall, the majority of the findings are exactly what one expects to find in sarcomas, with TP53, and RB mutations being among the most common findings. When the paper lists off strings of genes with mutations/deletions/amplifications, it is challenging to get a sense of what is novel or exciting, as many of these are familiar findings. I would urge the authors to find a way to highlight or emphasize to the reader more clearly what are the most novel or exciting/functional/therapeutic alterations they have elucidated here.

5. It is misleading to claim that “whole exome, transcriptome, and methylation data from 99 MFS” (Line 37) [was generated], when in fact only 41 cases underwent WES and methylation analysis, and 29 for RNA seq, while 54 cases only had targeted sequencing done.

6. In the methylation analysis, the authors have used skeletal muscle as a “normal control” and appear to have included these samples when performing hierarchical clustering with the MFS cases. These “controls” should be removed both from the text/figure and clustering analysis. Skeletal muscle is not a normal counterpart of MFS (which are, if anything, fibroblastic in differentiation) and the inclusion may be skewing the resulting clusters. I am curious as to the significance of the 3 MFS clusters. While the authors do a nice job in the supplemental tables elucidating the different genes and processes that are differentially methylated between clusters, methylation can also be

used to determine leukocyte fraction, and I wonder if there is any difference in leukocyte fraction between the 3 MFS clusters; as this may also contribute to the survival differences between clusters.

8. The conclusion that MFS are more like epithelial malignancies in terms of their genetic epigenetic profile comes out of the blue and is totally unsupported by any prior statements in the text. The pathways (e.g. cell cycle checkpoints, RAS/AKT pathway etc, are all known to be involved in sarcoma as well as carcinoma). MFS have a higher level of SCNAs than most carcinomas, but this is not delved into either. Moreover, the authors do not discuss how methylation profile is different than would be expected for a mesenchymal malignancy or the evidence for this statement. Finally, the logic by which 51% of the cases have actionable mutations is, at best wishful thinking given actual success rates with targeted therapies in sarcoma to date, and the fact that none of the citations for these agents address sarcoma clinical trials.

Minor

9. Given that this study has WES data on at least a portion of cases, I would be interested to see a characterization of telomere length in cases with ATRX mutations as a functional assessment of this findings (see Ding, Z., Mangino, M., Aviv, A., Spector, T., and Durbin, R. (2014)).

10. Please speculate on the functional significance of TET mutations? These are usually seen in hematologic malignancies – what role would the authors expect these are playing in MFS?

11. In many places, the text lacks supporting quantitative values or p-values for general statements made. A few instances include – eg. Line 169 needing specific #s for mdm2 in DDLPS, cdk6 in MFS/UPS; lines 247-249 lacks statistical values to back up general claims in text. It would strengthen the manuscript to include such values where possible.

12. Line 162 claims that 216 sarcomas were analyzed to create a detailed map of molecular alterations, but text is not much detailed or novel in the paragraph that follows.

13. Line 167 “unique mutational profile” needs more detail – how are these unique? Describe the differences and why they are meaningful.

14. Fig 2 and 3a and 4a are repetitive of figure 1. This data does not need to be shown twice

15. In fig 2a/2b it appears that the patients without cell cycle checkpoint alterations have improved survival compared to those that do. Perhaps, in addition to the individual genes shown in table S12, it would be helpful to do an outcome analysis of all 68% of cases with a checkpoint alteration vs the 32% without.

16. For the BRAF fusion gene, it would be helpful to report what is SLC37A and why this fusion results in overexpression of BRAF.

17. I do wonder if MFS 1 (being a superficial sarcoma) might represent a pleomorphic dermal sarcoma (with UV damage accounting for the high mutational rate)?

18. While data in supplemental show amplification of VGLL3, as has been noted in several other publications as a common finding in UPS/MFS, the authors opt not to highlight this locus on the GISTIC plot? Is there a reason for this?

19. I might argue that a q of 0.1 for significantly mutated genes is on the high side, but I can see that it was chosen in order to allow the authors to include CDKN2A/B in their list of SMGs.

Reviewers' comments:

Reviewer #1

Comment #1.

The authors should provide a flowchart to clarify the order in which different types of sequencing occurred and how many samples received each type of sequencing.

We appreciate your comment for better presentation of our analysis to the readers. We have provided this information as a Supplementary Figure 8 in addition to Supplementary Table 16.

Comment #2.

The authors describe generation of whole transcriptome data for 29 MFS samples, but provide analysis of only fusions gene identified. It would be interesting to see an analysis of any genes that were significantly upregulated or downregulated.

We appreciate your comment. We have performed additional analysis to identify significantly up-regulated and down-regulated genes among 29 MFS cases, and provided results in Supplementary Tables 8 and 9, respectively, according to the reviewer's suggestion. By this new analysis, we found that several interesting molecules were in the top-ranked genes, that include growth factors (*FGF8*, *BMP7*, *IGF2*, *TGFB2*, and *FGF18*), oncogene (*CTNNB1*), and immune-related molecules (*ICOS* and *TIGIT*). They could also be potentially novel targets for MFS. We have shown this information in Supplementary Figure 2 and Results section as below.

Page 13, Line 2

“Deregulated gene expression in MFS

We investigated significantly up-regulated and down-regulated genes in 29 MFSs (Supplementary Tables 8 and 9, respectively). The top-ranked genes included growth factors (*FGF8*, *BMP7*, *IGF2*, *TGFB2*, and *FGF18*), oncogene (*CTNNB1*), and immune-related molecules (*ICOS* and *TIGIT*) (Supplementary Table 8, Supplementary Figure 2). They could also be potentially novel targets for MFS. Although not in the top-ranked genes, previously reported prognostic genes including *CDK6*, *AMACR*, *SKP2*, *EZR*, and *MET*, were also significantly up-regulated in our cohort [22-26].”

Comment #3.

Line 37: The authors should reword the sentence, as it appears to suggest that 99 MFS samples underwent WES, but this is not the case based on later descriptions of sequencing.

We appreciate your comment. Our statement was misleading. We changed the sentence as suggested.

Page 3, Line 3

“Using 41 MFSs as a discovery set, we underwent whole exome sequencing (N=41), RNA sequencing (N=29), and methylation analysis (N=41). We subsequently performed targeted sequencing of 140 genes in the entire cohort of 99 MFSs to validate the results in the discovery samples. We also combined 17 MFSs data from TCGA to characterize the molecular features of MFS.”

Comment #4.

In supplementary table 1, the authors should clarify the way in which the q-value is produced.

We appreciate your comment. We described estimation method of significantly mutated genes and methods of p-value and the q-value calculations in Supplementary Methods (Page 2, Line 5-14). Also, we added this description in the Supplementary Tables 1 and 4 with some corrections according to your advice

It is unusual that so many of the q values are equal to one, and it unclear how this q-value relates to the p-value.

The q-value is False Discovery Rate (FDR) and calculated from the p-value using multiple testing adjustments. The multiple testing adjustments were performed using the method described by Benjamini and Hochberg, which is commonly used and the formula of the FDR is the following.

$$\text{FDR} = p \times N / i,$$

p: p-value of the gene,

N: the number of all genes (We used 20,075 genes for the evaluation),

i: the number of genes with p-value less than or equal to the p-value of the gene,

p x N: the number of false positive genes,

$p \times N / i$: the rate of false positive genes in genes with p-value less than or equal to the p-value of the gene.

Therefore, in the cases where N is large, if p is not low and/or i is small, FDRs become more than one. Since FDR is the probability, FDR is commonly showed as one for descriptive purposes when FDR is greater than one. When there are few significantly mutated genes, the cases where many of the q values are equal to one occur commonly and frequently as in other studies.

Comment #5.

In line 172 the authors should reword sentence, as TP53 is the most frequently gene encoding a protein involved in cell cycle regulation in their MFS sample.

We appreciate your comment. Based on the Comment #11 and #12 from Reviewer #2, we have rewritten in a more detailed manner throughout the section “Comparison of genetic alterations among soft tissue sarcomas with complex karyotypes” also by taking your comment (Page 10, Line 10).

Comment #6.

Minor spelling change: change to exclusive

We appreciate your comment. We checked all through the manuscript and changed as appropriate.

Comment #7.

The authors should clarify the selection criteria used for choosing the MFS samples that would undergo RNA-sequencing and/or methylation array analysis

We appreciate your comment. We selected samples for RNA sequencing simply by their RNA qualities for library construction. Samples that had insufficient quality of RNA (RIN < 6.5) were excluded for RNA sequencing as stated in the text (Page 26, Line 12). However, this process is complicated as you suggested, therefore, we provided the flowchart of the analysis as Supplementary Figure 8 for the reader to understand easily.

Comment #8.

The authors should provide information how the quantity of transcript for the SLC37A3-BRAF found in the MFS samples compares to V600E BRAF samples, as the western blot appears to show significantly more SLC37A3-BRAF protein when compared to the V600E lane.

We appreciate your comment. We analyzed transcripts in stable NIH3T3 clones by RT-PCR and quantitative real-time PCR (qPCR) as shown in the figures below. Expression of BRAF-V600E mutant and SLC37A-BRAF fusion was detected by specific RT-PCR, respectively (panel a). qPCR analysis using UPL probe on the common BRAF region between BRAF-V600E mutant and SLC37A-BRAF fusion identified overexpression of the

fusion compared to the V600E mutant (panel b). Overexpression and/or stability of the fusion transcript might contribute to abundant protein expression.

Method: Total RNA (1 ug) was reverse-transcribed into cDNA with iScript (Bio-Rad, CA). cDNA was subjected to PCR amplification with GXL-Taq (Takara Bio, Japan) and primer sets of BRAF-008U20 (CACTTCCGGAGGAGGGGGT) and BRAF-340L21 (AACGGTATCCATTGATGCAGA) for BRAF-V600E, SLC37A-CF1 (CTGCTCACTTCTTCAGTTATTCGTT) and BRAF-CR1 (CTAGCTTGCTGGTGTATTCTTCATAG) for SLC37A3-BRAF fusion, and mGAPDH-CF1 (AGCTTGTCATCAACGGGAAG) and mGAPDH-CR1 (CCTGCTTACCACCTTCTTG) for mouse Gapdh. The expression of the human BRAF transcripts were assayed by qPCR using the LC96 thermal cycler (Roche, Germany). BRAF expression was normalized to mouse Gapdh expression. Primers used for qPCR are as follows: human BRAF (Fwd-ATCCCAGAGTGCTGTGCTG, Rev-GGAAATATCAGTGTCCCAACCA, UPL probe #58), mouse Gapdh (Fwd-TGTCCGTCGTGGATCTGAC, Rev-CCTGCTTACCACCTTCTTG, UPL probe #80).

Comment #9.

An effort should be made to identify any samples that were part of both methylation array and RNA seq experiments. Identifying highly methylated genes that have correspondingly reduced transcription levels would help validate methylation results and remove some noise from the methylation analysis.

We appreciate your comment. We compared the methylation and expression data in 29 tumor samples and correlation was evaluated using Spearman's correlation coefficient. Genes with significant correlations were summarized in Supplementary Table 14. In most genes, promoter methylation and expression were negatively correlated, confirming little noise in our methylation analysis. This analysis also revealed that promoter methylation silenced a wide range of genes including those associated with development, cell cycle and cell adhesion regulators, and homeobox genes. We have shown this information in the Results section (Page 15, Line 7).

**Reviewer #2
Comment #1.**

The biggest issue I have with this study is the uncertainty I have over the actual diagnosis of the cases included in this study as MFS. The diagnosis of MFS can be challenging and controversial, and it is unclear which criteria were used to diagnose the tumors for this study. While the WHO was cited, it would be helpful to have some specific criteria outlined in the methods. I am deeply dubious about the diagnosis of the cases which later are reported to have *MDM2* amplifications, and suspect these to be dedifferentiated liposarcoma (DDLPS). These should be excluded from the study or included as reclassified DDLPS in table S6. Please describe the pathology review that was performed for case inclusion and diagnostic inclusion/exclusion criteria for these cases – simply citing the WHO is insufficient here.

We appreciate this very important comment on the difficulty in analyzing sarcoma cohorts. All the cases in our study were carefully reviewed by a pathologist (AY) with expertise in soft-tissue tumors. We diagnosed MFS for pleomorphic spindle cell (or rarely epithelioid-appearing) sarcoma of uncertain/undifferentiated phenotype showing stromal myxoid change that accounted for at least 10% of the total tumor volume, according to WHO and Mentzel et al (Mentzel et al, Am J Surg Pathol 1996;20(4):391-405). Any evidence of unequivocal differentiation toward specific lines was considered incompatible with MFS. For example, the presence of lipoblasts, rhabdomyoblasts, coexisting neurofibroma, coexisting well-differentiated liposarcoma components excluded the diagnosis of MFS. However, myofibroblastic differentiation represented by amphophilic fibrillary cytoplasm and the immunorexpression of actin and/or desmin (with no accompanying expression of myogenin or caldesmon) was accepted for the MFS designation. MFS mimics such as myxoinflammatory fibroblastic sarcomas and pleomorphic dermal sarcomas were excluded. Curvilinear vessels in the myxoid areas and diffuse infiltration along fascial planes were characteristic of MFS; however such features were not absolute requirements for the diagnosis. Of note, *MDM2* amplification status was not considered in the review process, unless the tumor was located in the internal trunk. Although *MDM2* amplification in pleomorphic sarcoma in the internal trunk is accepted by most as probable dedifferentiated liposarcoma with unsampled or overgrown well-differentiated liposarcoma elements, the classificatory significance of *MDM2* amplification in peripheral sarcomas (including those arising in the trunk wall) is very controversial. We are aware of a view that equates such sarcomas with dedifferentiated liposarcomas (Le Guellec et al, Am J Surg Pathol 2014;38(3):293-304), but the data are limited and it is still premature for such a view to direct our diagnosis. In daily practice, one could still diagnose *MDM2*-amplified MFS at the peripheral site in the absence of synchronous or metachronous well-differentiated liposarcoma histology, with a note regarding the abovementioned nosologic controversy. In this study, we therefore maintain *MDM2* amplified tumors within MFS cohort. Parenthetically, all such cases in this study lacked *CDK4* amplification, in contrast to most (~80%) dedifferentiated liposarcomas that coamplify *MDM2* and *CDK4* (Asano N et al, Oncotarget 2017;8(8):12941-12952), and this might suggest some genetic difference.

Also, it would be important to note whether these tumors are primary, recurrent, metastatic disease, or if the specimens were previously subjected to radiation or chemotherapy as neoadjuvant therapy and disease status may affect the findings. I note that the text does specify that WES and methylation studies included only

primary, but this should be stated for all cases. (The only place the tumor status appears is in supplemental table 13 – did you see any difference in mutations between primary and metastatic/recurrent tumors).

We appreciate your comment. According to your suggestion, we added the following sentence in the Patients and samples section.

Page 23, Line 12

“All tumor samples in the discovery set (N=41) and TCGA dataset (N=17) were from primary tumors. Tumor samples in the validation set (N=58) included 38 primary tumors, 17 recurrent tumors, and 3 metastatic specimens (Supplementary Figure 8).”

Also, information on previous chemotherapy or radiotherapy was provided in Supplementary Table 16.

According to your suggestion, we analyzed the difference in mutation profile of major genes among the sample status. No genes were detected as frequent alterations in metastatic samples. However, frequency of somatic mutation of *GNAS* (primary 0.0%, metastatic 0.0% vs. recurrent 17.6%; $p < 0.001$, chi-squared test) and *SETD2* (primary 1.0%, metastatic 0.0% vs. recurrent 11.8%; $p = 0.036$, chi-squared test) were significantly higher in recurrent specimens compared with primary tumors. We have added this information in Results section as below.

Page 8, Line 9

“We compared mutation frequencies of these driver genes among the sample status: primary (N=96), recurrent (N=17), and metastatic (N=3). None was statistically more frequent in metastatic samples. However, frequencies of *GNAS* (primary 0.0%, metastatic 0.0% vs. recurrent 17.6%; $p < 0.001$, chi-squared test) and *SETD2* (primary 1.0%, metastatic 0.0% vs. recurrent 11.8%; $p = 0.036$, chi-squared test) mutations were significantly higher in recurrent specimens.”

Comment #2.

The 25 “MFS” cases the authors have used from the TCGA dataset have not been subjected to any sort of QC protocol for this study. I am aware that the curated TCGA dataset will not be available until after that manuscript is published, nevertheless, nearly half of these 25 cases have poor quality data (high level of artifacts) or should not be considered MFS after adequate pathology review.

We appreciate your comment. Based on the published TCGA paper, we included 17 certified MFS cases from TCGA cases in our study, and revised all data related to this change. We analyzed the molecular data of 17 MFS cases obtained from TCGA dataset based on the sample IDs of MFS in the TCGA sarcoma manuscript (Abeshouse et al, Cell 2017;171:950-965) because these cases were subjected to sequence QC and pathologic review as described in their paper. We also cited this manuscript as Reference 10.

Comment #3.

I am concerned about the methodology used to determine amplification/deletion in tumors with targeted sequence data only (which represent the majority of cases in the

study), as well as the definition of “homozygous deletion” for both WES and targeted sequence data. In the methods, it states that tumors were assumed to be diploid when determining copy number. In my experience, this is not a safe assumption to make in MFS or any other high grade complex karyotype sarcoma. I would strongly urge the authors to re-evaluate their methodology here.

Also, the definition of homozygous deletion is imprecise. Line 378 “ratio of tumor read depth to normal read depth was <0.50 ” – this could also be low purity even with a low level deletion. I would accept that this methodology would catch heterozygous deletions, but am not convinced it is sufficient to determine these to be homozygous.

Line 380 “deletion affecting a known driver gene”. There is no reason to presume that a deletion affecting a known driver gene has to be homozygous. Tumor suppressors are frequently bi-allelically mutated and deleted. The use of circular terminology in defining deletions here minimizes the fact that heterozygous deletions are at least as common as homozygous in these tumors, yet nowhere are heterozygous deletions mentioned or displayed in this work.

We appreciate your comments regarding our copy number analysis. As per the reviewer’s comments, we clarified or corrected three points described below in the revised manuscript.

1. Methodology to determine amplification/deletion in tumors with targeted sequence data only

We agree with the reviewer that sarcoma genomes frequently show complex karyotypes, and our assumption that it would be diploid for evaluating copy number changes was inappropriate. The accuracy of the calculation of the ploidy depends on the size of target gene regions. Compared to whole exome sequencing (WES) analyzing about 20,000 genes, target sequencing (TS) covers only 140 genes. Therefore, the number of observed heterozygous SNPs and somatic mutations is small and it makes calculations of ploidy very difficult in TS cases.

Previous studies using array-based CGH commonly applied copy number (CN) ratio between tumor and normal genomes. When CN ratio is used, the aberrant number of individual chromosomes can be reflected in the CN ratio of each individual chromosome. Therefore, we changed our methods and used CN ratio for evaluating copy number changes in samples with target sequencing data only (Page 26, Line 4-9).

2. The definition of homozygous deletion (HD) (Line 378)

We are sorry that the description of our method to detect HD was confusing. We first selected genes whose adjusted CN ratio was ≤ 0 (see step 2 in the revised method below). Therefore, we did not include heterozygously deleted genes for detecting HD. However, as described above, we had difficulty in evaluating tumor purity in cases with small number of observed somatic mutations, such as in cases with target sequencing data only, that would include false positives. Therefore, we further selected genes with additional filtering conditions shown in step 3.

We believe that this procedure is conservative because at first we selected HDs using the true condition of “CN ratio adjusted by tumor purity ≤ 0 ” and then removed false positive

candidates using the preliminary condition of “CN ratio before adjustment by tumor purity < 0.5”. The reason why we used the threshold of 0.5 is the following.

Adjusted CN ratio (AR) = $1 - (1 - R) / TP$; where

R: CN ratio before adjustment,

TP: tumor purity.

In the case of HDs, AR = 0 and R = 1 – TP.

If TP = 0.75, R = 0.25. So if R < 0.25, we cannot detect HDs with TP < 0.75.

If TP = 0.50, R = 0.50. So if R < 0.50, we cannot detect HDs with TP < 0.50.

If TP = 0.25, R = 0.75. So if R < 0.75, we cannot detect HDs with TP < 0.25.

Since the average of TP of MFS is about 0.5 based on our WES data, the condition of “R < 0.25” will have many false negatives. Therefore, we determined the condition of “R < 0.50” could have less false negatives and false positives compared to other conditions. The condition of “R = 0.5” does not represent heterozygous in this filtering. It represents the CN ratio before adjustment when AR = 0 and TP = 0.5.

We modified the sentence and added detailed explanations in the methods to clearly show the detailed definition of HD as follows.

Page 25, Line 1-9

- (1) First, we identified genes with statistically significant CN change by GISTIC analysis. Next, we determined whether these genes located in the region of HD under the following severe conditions.
- (2) CN ratio adjusted by tumor purity ≤ 0
- (3) When the number of somatic mutations is small, it is difficult to estimate accurately the tumor purity, and the adjusted CN ratio may not be accurate in some cases. So we removed genes from HD candidates when the CN ratio before adjustment is greater than 0.5. We think our method is conservative because at first we selected HDs using the true condition of “CN ratio adjusted by tumor purity ≤ 0 ” and next as a precaution we removed false positive candidates using the preliminary condition of “CN ratio before adjustment by tumor purity < 0.5” to decrease both false negatives and false positives.

3. Line 380 “deletion affecting a known driver gene”.

As the reviewer pointed out, there are several ways to inactivate tumor suppressor genes, such as double mutations, one mutation + loss of heterozygosity (LOH) (with or without copy loss), and homozygous deletion. We have included all nucleotide substitutions such as missense or nonsense mutations to evaluate driver genes in our analysis. Therefore, in addition to HD, other modes of two hits should be included in our analysis. However, as the reviewer commented, we did not describe about frequent copy loss regions in detail. We performed GISTIC analysis to isolate significant copy loss regions, and identified 14 regions for significant copy number loss ($q < 0.01$) that includes heterozygous and homozygous deletion. We have modified the description to make this important finding clearer to the readers as below.

“Using GISTIC 2.0 analysis, significant focal amplifications ($q < 0.25$) and copy number losses (homozygous and heterozygous deletions) ($q < 0.01$) were identified at 29 loci (15

amplifications and 14 copy number losses), including known oncogenes (*JAK1*, 1p31.3; *VGLL3*, 3p11.2; *CCND1*, 11q13.2; *SYK*, 9q21.2; *FOXA1*, 14q11.2; *NKX2-1*, 14q11.2; *KRAS*, 12q12; and *CDK6*, 7q21.3) (Supplementary Figure 1a, Supplementary Table 2), and tumor suppressors (*TP53*, 17p13.1; *MST1R*, 3p21.31; *CDKN2A* and *CDKN2B*, 9p21.3; *RBI*, 13q14.2; and *CDHI*, 16p11.2) (Supplementary Figure 1b, Supplementary Table 3).”

We did not include heterozygous deletion data for evaluating driver genes because copy loss covers many genes and it is difficult to isolate target gene by copy number data alone. We listed known cancer associated genes as final candidates for HD in Supplementary Table 4.

Comment #4.

Overall, the majority of the findings are exactly what one expects to find in sarcomas, with TP53, and RB mutations being among the most common findings. When the paper lists off strings of genes with mutations/deletions/amplifications, it is challenging to get a sense of what is novel or exciting, as many of these are familiar findings. I would urge the authors to find a way to highlight or emphasize to the reader more clearly what are the most novel or exciting/functional/therapeutic alterations they have elucidated here.

We appreciate your comment. MFS is one of the common histological subtypes of adult soft tissue sarcoma with complex karyotype and has characteristic clinical presentations including the infiltrative growth pattern and high propensity for local recurrence. To our knowledge, the largest comprehensive genetic analysis for MFS ever reported was published in 2010 as we cited as Reference 7. That study analyzed only 38 cases and the comprehensive molecular alterations and diversity of MFS have remained to be unknown.

Our study collected and analyzed the largest MFS cohorts and provided two main discoveries as follows.

1. Molecular genetic architecture of MFS

By integrating somatic substitutions and copy number changes, we identified 14 driver genes with significant q value, most of which have not been identified as driver genes in MFS to date including *CCND1*, *KRAS*, *WNT11*, *CDK6*, *GNAS*, *FOXA1*, *NKX2-1*, *SYK*, and *JAK1*, and some of them are potential therapeutic targets. Moreover transcriptome analysis revealed a novel *BRAF* fusion genes and up-regulated genes that would be novel therapeutic targets.

2. Prognostic and immune-related molecular profiling of MFS

Using genetic and DNA methylation profiling, we characterized unique molecular subclasses in MFS. These subclasses were also associated with patients’ prognosis and different immune-microenvironments (as shown in our response to comment #6 below).

We have rewritten the conclusions to make the most novel and important findings clear to the readers as below.

Page 21, Line 15

“In conclusion, our comprehensive and integrative molecular analyses of MFS have uncovered substantial recurrent driver genes, most of which have not been identified as driver genes in MFS to date. DNA methylation patterns clustered into three subtypes closely

associated with immune cell compositions, especially the fraction of CD8+ T cell, as well as unique combinations of driver mutations and prognosis. The driver alterations, significantly up-regulated genes, immune cell compositions and DNA methylation pattern identified in this study, would help to identify potential therapeutic targets as well as molecular subtypes with prognostic significance. These results are expected to provide a valuable basis for the development of precision medicine approaches, including molecular diagnostics and therapeutics, in MFS.”

Comment #5.

It is misleading to claim that “whole exome, transcriptome, and methylation data from 99 MFS” (Line 37) [was generated], when in fact only 41 cases underwent WES and methylation analysis, and 29 for RNA seq, while 54 cases only had targeted sequencing done.

We appreciate your comment and totally agree with your opinion. We changed the sentence as below and inserted Supplementary Figure 8 for clarity.

Page 3, Line 3

“Using 41 MFSs as a discovery set, we underwent whole exome sequencing (N=41), RNA sequencing (N=29), and methylation analysis (N=41). We subsequently performed targeted sequencing of 140 genes in the entire cohort of 99 MFSs to validate the results in the discovery samples. We also combined 17 MFSs data from TCGA to characterize the molecular features of MFS.”

Comment #6.

In the methylation analysis, the authors have used skeletal muscle as a “normal control” and appear to have included these samples when performing hierarchical clustering with the MFS cases. These “controls” should be removed both from the text/figure and clustering analysis. Skeletal muscle is not a normal counterpart of MFS (which are, if anything, fibroblastic in differentiation) and the inclusion may be skewing the resulting clusters.

We appreciate your comment. As the reviewer commented, the cell of origin of MFS is controversial, and we agree with the reviewer's suggestion that skeletal muscle may not be a normal counterpart of MFS. Based on the comment, we re-performed hierarchical clustering using only tumor samples by excluding normal skeletal muscle data. As a result, 41 MFSs were clustered into three subgroups and each cluster was more clearly correlated with major genetic alterations. Cluster A was characterized by *MDM2* amplification and alterations of genes associated with histone/chromatin modification. Cluster B was characterized by *RBI* and *ATRX* alterations. Cluster C was characterized by *CDKN2A/B* and *NF1* alterations. Also, these clusters were significantly associated with overall survival. We changed the Figure 6 and corresponding text and legends accordingly as shown below.

Results: Page 13, Line 14-Page 15, Line 4

Methods: Page 27, Line 18

Figure legends: Page 38, Line 7

I am curious as to the significance of the 3 MFS clusters. While the authors do a nice job in the supplemental tables elucidating the different genes and processes that are differentially methylated between clusters, methylation can also be used to determine leukocyte fraction, and I wonder if there is any difference in leukocyte fraction between the 3 MFS clusters; as this may also contribute to the survival differences between clusters.

We very much appreciate for this quite exciting suggestion and comment. To independently and fully evaluate immune cell composition among methylation-based subclasses, we inferred the differences in immune cell fraction from RNA sequencing data using CIBERSORT (Newman et al, Nat Methods 2015;12(5):453-457). As the reviewer speculated, there was significant difference of immune cell composition among three clusters. Especially, the average fraction of CD8+ T cell was significantly higher in Cluster A that showed better prognosis compared to others (P = 0.033, Figure 6, Supplementary Figure 4). This result suggested high fraction of CD8+ T cell may be associated with better outcome in MFS as the reviewer speculated. We included this new data in the Results (Page 15, Line 14) and Discussion (Page 21, Line 1).

Comment #7.

The conclusion that MFS are more like epithelial malignancies in terms of their genetic epigenetic profile comes out of the blue and is totally unsupported by any prior statements in the text. The pathways (e.g. cell cycle checkpoints, RAS/AKT pathway etc, are all known to be involved in sarcoma as well as carcinoma). MFS have a higher level of SCNAs than most carcinomas, but this is not delved into either. (*Moreover, the authors do not discuss how methylation profile is different than would be expected for a mesenchymal malignancy or the evidence for this statement.)

We appreciate your comment. Our conclusion that MFS is more like epithelial malignancies in terms of their genetic profile was not clearly supported by the text and may be misleading because, as the reviewer commented, the pathways including cell cycle checkpoints or RAS/AKT pathway are known to be involved also in sarcoma, represented by dedifferentiated liposarcoma, as well as carcinoma. We should discuss and conclude based on objective findings of the study, therefore, we modified the conclusion section as below.

Page 21, Line 15

“In conclusion, our comprehensive and integrative molecular analyses of MFS have uncovered substantial recurrent driver genes, most of which have not been identified as driver genes in MFS to date.”

We did not analyze epigenetic similarity between epithelial malignancies and MFS in the text. This would be quite interesting analysis, however, we need a larger collection of methylation data from both epithelial and mesenchymal tumors, that would be beyond the scope of this study.

Finally, the logic by which 51% of the cases have actionable mutations is, at best wishful thinking given actual success rates with targeted therapies in sarcoma to date, and the fact that none of the citations for these agents address sarcoma clinical trials.

We appreciate your comment. As the reviewer commented, due to the rarity and heterogeneity, clinical trials focusing on soft tissue sarcoma have not been sufficiently conducted to date. As far as we surveyed, there is no publication that describes any clinical trial of MFS or soft tissue sarcoma with driver alterations identified in our study. To conquer this difficulty, basket trials that target a specific biomarker/molecular alteration regardless of cancer types could be more feasible and some trials have already been ongoing. Therefore, we can only conclude driver alterations identified in this study have a potential to be druggable, which is also the case in other studies such as Reference 7 (Barretina et al, Nat Genet 2010;42(8):715-721). Based on the reviewer's comment, we have rewritten the conclusion in a more unassuming fashion as below.

Page 22, Line 1

“The driver alterations, significantly up-regulated genes, immune cell compositions and DNA methylation pattern identified in this study, would help to identify potential therapeutic targets as well as molecular subtypes with prognostic significance. These results are expected to provide a valuable basis for the development of precision medicine approaches, including molecular diagnostics and therapeutics, in MFS.”

Comment #8.

Given that this study has WES data on at least a portion of cases, I would be interested to see a characterization of telomere length in cases with ATRX mutations as a functional assessment of this findings (see Ding, Z., Mangino, M., Aviv, A., Spector, T., and Durbin, R. (2014)).

We appreciate your comment. We totally agree that functional assessment of the *ATRX* mutations by estimating telomere length is interesting. However, we performed only whole exome sequencing (WES) (capturing only coding exons), not whole genome sequencing (WGS), and our WES did not capture the telomere region which is a non-coding genome region. Recent TCGA paper (Reference 10) certainly applied the TelSeq algorithm (Ding et al, Nuclei Acids Res 2014;42(9):e75), which is also suggested by the reviewer, to WES data by carefully evaluating analytical conditions using WGS data as a positive control.

Although WGS data to validate the TelSeq estimation based on WES data were unavailable in our WES samples, we applied WES data of 41 MFS cases to TelSeq and calculated the telomere length. We then investigated the association of *ATRX* mutations with telomere length in our cohort having 4 *ATRX* mutant cases and 37 wild-type cases. We found the tendency toward longer telomere length in *ATRX* mutant cases as shown in the panel below, however, it was not statistically significant ($P = 0.483$).

The reasons why we could not find statistically significant difference would be partly attributable to the low statistical power due to the small sample size, or the presence of other genetic or epigenetic alterations that also affected telomere length in MFS. In addition, we would like to emphasize that we should interpret the result carefully because we did not have WGS data of our samples for validating the TelSeq outputs, and accuracy of telomere length estimation by this analysis was not guaranteed at present data set. Indeed, the off-target region of our WES is about 20% and the off-target region for our 100x of WES corresponds about only 1x of WGS. TelSeq's paper (Ding et al, Nuclei Acids Res 2014;42(9):e75) indicated that sequence reads corresponding to 2.5x or more of WGS are required for an accurate estimate. We would like to analyze the accurate relationship between *ATRX* mutations and telomere length using WGS data in the future study.

Comment #9.

Please speculate on the functional significance of TET mutations? These are usually seen in hematologic malignancies – what role would the authors expect these are playing in MFS?

We appreciate your comment. TET2 is a member of the TET family of proteins and known to be frequently mutated in a wide variety of hematopoietic tumors. Also *TET2* mutations have been reported in solid cancer including renal cancer (5.7%) (Sato et al, Nat Genet 2013;45(8):860-867) and colorectal cancer (2.3%) (Cancer Genome Atlas Network, Nature 2012;487(7407):330-337), but not in sarcoma. TET2 protein catalyzes the conversion of 5-methylcytosine to 5-hydroxymethylcytosine, which is a critical step in DNA demethylation. However, the accurate mechanism of *TET2* alterations contributes to tumorigenesis remains to be unknown in solid cancer. All four *TET2* mutations detected in MFS were nonsense mutations that result in truncating cysteine-rich or catalytic domains and lose DNA demethylating activity. Therefore, we speculate that inactivation of TET2 in MFS may lead to subsequent changes in DNA methylation patterns and contributed to the tumorigenesis.

Consistently with this speculation, we found that all *TET2* mutated cases were in the Cluster A with unique DNA methylation pattern.

Comment #10.

In many places, the text lacks supporting quantitative values or p-values for general statements made. A few instances include – eg. Line 169 needing specific #s for *mdm2* in DDLPS, *cdk6* in MFS/UPS; lines 247-249 lacks statistical values to back up general claims in text. It would strengthen the manuscript to include such values where possible.

We totally agree with your suggestion. We added the quantitative values or p values as appropriate.

Comment #11.

Line 162 claims that 216 sarcomas were analyzed to create a detailed map of molecular alterations, but text is not much detailed or novel in the paragraph that follows.

Comment #12.

Line 167 “unique mutational profile” needs more detail – how are these unique? Describe the differences and why they are meaningful.

We appreciate your comments. The section “Comparison of genetic alterations among soft tissue sarcomas with complex karyotypes” seems a little bit abstract as you suggested, and need to be explained more concretely. We have rewritten in a more detailed manner throughout this section as described below.

Page 10, Line 11

“Although previous studies have partially revealed the importance of the p53 signaling pathway, cell cycle regulators, and the RTK-RAS-PI3K pathway in soft tissue sarcomas with complex karyotypes, such as MFS, dedifferentiated liposarcoma (DDLs), LMS, and UPS [7, 18], a comprehensive overview of the molecular pathogenesis of these tumors, including differences among histological subtypes, has not previously been elucidated. Therefore, to construct a detailed map of the molecular alterations across diverse sarcoma subtypes, we analyzed data from an additional 174 samples, including three major histologic subtypes of soft tissue sarcoma with complex karyotypes (50 DDLs, 80 LMSs, and 44 UPSs), in addition to the 116 MFSs. The frequency of major genetic alterations in each subtype is provided in Supplementary Table 6. Although genes that belong to p53 signaling pathway, cell cycle regulators, and the RTK-RAS-PI3K pathway were widely affected in these sarcomas, each histological subtype exhibited a unique combination of mutation profile. In p53 signaling pathway, DDLs was distinctive in that *MDM2* was more frequently affected (94.0%, $p < 0.001$) whereas it was not the case for other three subtypes. In addition, high incidence of amplification of *CDK4* (86.0%, $p < 0.001$) made DDLs a more distinct entity compared to others in which *RBI* and *CDK6* were more frequently affected. In the RTK-RAS-PI3K pathway, *PTEN* was predominantly affected in LMS (18.8%, $p < 0.001$) whereas *NF1* and *KRAS* mutations were rare. By contrast, these two were predominantly seen in DDLs/MFS/UPS and DDLs/MFS, respectively. *NKX2-1* alterations were exclusively found in MFS ($p = 0.013$).”

Comment #13.

Fig 2 and 3a and 4a are repetitive of figure 1. This data does not need to be shown twice.

We appreciate your comment. However, we believe that it is not easy to understand the mutually exclusive profiles for readers only by Figure 1 and we would like to leave the figures to emphasize mutually exclusive profiles by sorting genes in these pathways.

Comment #14.

In fig 2a/2b it appears that the patients without cell cycle checkpoint alterations have improved survival compared to those that do. Perhaps, in addition to the individual genes shown in table S12, it would be helpful to do an outcome analysis of all 68% of cases with a checkpoint alteration vs the 32% without.

We appreciate your comment. According to the reviewer's suggestion, we further performed survival analysis and found that overall survival of patients with cell cycle check point alterations was significantly poorer compared with those without alterations ($p = 0.001$). We added Kaplan-Meier survival plots stratified by cell cycle checkpoint alterations as Supplementary Figures 5m and 6m.

Comment #15.

For the BRAF fusion gene, it would be helpful to report what is SLC37A and why this fusion results in overexpression of BRAF.

We appreciate your comment. SLC37A3 is a solute carrier family 37, member 3 protein that has a transmembrane transporter activity and lacks putative protein-dimerization domain such as coiled-coil or zinc-finger domains.

We speculated that over expression of SLC37A-BRAF fusion protein depends on the promoter activity of *SLC37A3*, as reported in other BRAF fusions such as *KIAA1549-BRAF*. We examined SLC37A3 expression across different tissues and cell lines in public databases. SLC37A3 gene expression is specifically high in uterus, cervix, and fibroblast (GTEx Portal, <https://www.gtexportal.org/home/>) and mesenchymal stem cell (BioGPS, <http://biogps.org/#goto=welcom>). By contrast, BRAF gene expression is low in fibroblast and mesenchymal stem cell. Although cell of origin of MFS remains to be unknown, this suggests that promoter swapping between *BRAF* and *SLC37A3* by structural rearrangement may increase expression of *SLC37A-BRAF* fusion gene compared to that of wild-type *BRAF*. We described this statement in Discussion section as below.

Page 20, Line 8

“In addition, we identified a novel *BRAF* driver gene fusion, *SLC37A-BRAF*, which could be targeted with anti-BRAF therapies [19-21]. SLC37A3 is a solute carrier family 37, member 3 protein that has a transmembrane transporter activity and lacks putative protein-dimerization domain such as coiled-coil or zinc-finger domains. We speculated that over expression of SLC37A-BRAF fusion protein depends on the promoter activity of *SLC37A3* because *SLC37A3* gene expression is specifically high in uterus, cervix, and fibroblast (GTEx Portal [53]) and mesenchymal stem cell (BioGPS [54]) whereas *BRAF* gene expression is low in fibroblast and mesenchymal stem cell. Although cell of origin of

MFS remains to be unknown, this suggests that promoter swapping between *BRAF* and *SLC37A3* by structural rearrangement may increase expression of *SLC37A-BRAF* fusion gene compared to that of wild-type *BRAF*.”

Comment #16.

I do wonder if MFS 1 (being a superficial sarcoma) might represent a pleomorphic dermal sarcoma (with UV damage accounting for the high mutational rate)?

Pleomorphic dermal sarcoma could indeed be histologically indistinguishable from MFS, with the location being the key discriminating factor. MFS1 was located in the subcutis of the lower extremity, and therefore, incompatible with the diagnosis of pleomorphic dermal sarcoma.

Comment #17.

While data in supplemental show amplification of *VGLL3*, as has been noted in several other publications as a common finding in UPS/MFS, the authors opt not to highlight this locus on the GISTIC plot? Is there a reason for this?

We appreciate your comment. We reformatted the Supplementary Figure 1 to include *VGLL3*. Also, we included *VGLL3* into Figures 6b and changed the text accordingly (Page 7, Line 3).

Comment #18.

I might argue that a q of 0.1 for significantly mutated genes is on the high side, but I can see that it was chosen in order to allow the authors to include *CDKN2A/B* in their list of SMGs.

We think that a q of 0.1 for significantly mutated genes is not on the high side and it was not chosen to include *CDKN2A/B*. In our previous studies (Totoki et al, Nat Genet 2014;46(12):1267-1273, Nakamura et al, Nat Genet 2015;47(9):1003-1010, Yachida et al, Cancer Cell 2016;29(2), 229-240) and also in studies from other groups (for example, most papers of TCGA and ICGC), a q of 0.1 for significantly mutated genes was applied.

The q-value is the False Discovery Rate (FDR) and was calculated from the p-value using multiple testing adjustments. The multiple testing adjustments were performed using the method described by Benjamini and Hochberg, which is commonly used and the formula of the FDR is the following.

$$\text{FDR} = p \times N / i,$$

p: p-value of the gene,

N: the number of all genes (We used 20075 genes for the evaluation),

i: the number of genes with p-value less than or equal to the p-value of the gene,

p x N: the number of false positive genes,

p x N / i: the rate of false positive genes in genes with p-value less than or equal to the p-value of the gene.

Therefore, the FDR is greater than the p-value and in the cases where N is large, the FDR increases from the p-value to a large degree. For example, in the case of *CDKN2A* in

Supplementary Table 4, the q-value of $6.33e-4$ in the AMM method increased from the p-value of $3.15e-7$. A FDR of 10% is not high. We think that the threshold of p-value of 0.01 or 0.05 and that of q-value of 0.1 or 0.2 are usually used. We are sorry that the Supplementary Table 4 had some skips, so you got confusing. We replaced the Supplementary Table 4 without skips. The method of the estimation of the significantly mutated genes and the methods of the calculations of the p-value and the q-value have described in Supplementary Methods (Page 2, Line 5).

Reviewer #1 (Remarks to the Author):

I thank the authors for addressing all of my concerns in the revised manuscript.

Reviewer #2 (Remarks to the Author):

This revised manuscript and methods is significantly improved from the original submission.

However, I continue to take issue with the inclusion of MDM2 amplified tumors in this study, and would prefer to see what I would consider to be a “cleaner” and less controversial data set excluding these 7 cases. Many academic centers worldwide use MDM2 amplification (in the appropriate morphologic and clinical context) to support a diagnosis of DDLPS in the extremities as well as the abdominal cavity.

Otherwise, I only have a few very minor additional comments/queries, below.

I am curious about the case with the BRAF gene fusion. BRAF fusions have been reported in myxoinflammatory fibroblastic sarcoma, which can mimic MFS. Do the authors think that this is a mis-classified case, or, instead, another example of shared molecular features (such as amplification of VGLL3 in a subset of both entities)?

In the up- and down-regulated gene list – what is the basis for comparison? (median of all genes?)

In the TCGA data, most sarcoma types with hypermethylation had worse outcomes than hypomethylated tumors – I would be curious as to what the authors speculate could account for their findings that the opposite was true in their series of MFS?

Some of the supplemental tables have different numbering on the excel tab than the file was submitted as (i.e table S13 the excel tab labels it as S12 though the heading says S13).

Also consider for simplicity's sake submitting several of the related tables as one excel file with tabs (assuming journal guidelines allow) – i.e Tables S10-13 could be tabs A-D in one table.

AS before, I am unclear on why the authors chose to reiterate the same data shown in figure 1 as 2a, 3a and-4a – It isn't that hard to follow; I would rather see these removed, condense the pathway panels into one figure and some of the more interesting findings from the supplemental data moved to main figures (such as the immune analysis for the methylation clusters). But I agree this may be a personal preference of mine, and others may feel differently.

I also wonder why the authors chose only to mention one rare fusion (albeit one involving BRAF), when there are several other recurrent fusions listed in the supplemental table that might be of interest despite not involving a known oncogene. While I acknowledge that generating cell models such as was done for the BRAF fusion for these fusions are beyond the scope of this paper, It would have been nice to include some speculation about these recurrent fusions and their functionality, if any.

Reviewers' comments:

Reviewer #1

Comment #1.

I thank the authors for addressing all of my concerns in the revised manuscript.

We appreciate your favorable and satisfied comment on our revised manuscript.

Reviewer #2

Comment #1.

This revised manuscript and methods is significantly improved from the original submission. However, I continue to take issue with the inclusion of MDM2 amplified tumors in this study, and would prefer to see what I would consider to be a “cleaner” and less controversial data set excluding these 7 cases. Many academic centers worldwide use MDM2 amplification (in the appropriate morphologic and clinical context) to support a diagnosis ofDDLPS in the extremities as well as the abdominal cavity.

We appreciate your comment. We are fully aware that some academic centers consider *MDM2* amplification as the evidence of dedifferentiated liposarcoma (DDLPS) even without well differentiated liposarcoma (WDLPS) component at the peripheral sites, but this is a controversial field where approaches can be different amongst pathologists. Although *MDM2* amplification is a good marker for DDLPS, there is evidence that it can also occur, albeit uncommonly, in non-DDLPS soft-tissue tumor types (e.g., Wallander ML et al, Arch Pathol Lab Med 2012;136(1):95-99). Because myxofibrosarcoma lacks entirely specific histological and molecular features, in the absence of WDLPS component, whether such *MDM2* amplified tumors belong to DDLPS category tends to result in a circular argument. In this regard, we would like to reiterate that none of our MFS cases harbored *CDK4* coamplification, unlike most dedifferentiated liposarcomas (92%) reported in TCGA.

Although we agree that excluding *MDM2* amplified subset would produce “cleaner” cohort, we are afraid, at the same time, such a selection may introduce a bias that could hamper us to capture the comprehensive picture of MFS. In addition, we would like to emphasize no significant difference in distribution of genetic alterations was noted when compared *MDM2* amplified and non-amplified MFS in our series. We would therefore like to maintain this small subset within the study cohort, and add discussion dedicated to this classificatory issue revolving around *MDM2* amplification to raise appropriate attention from the readership (Page 19, Line 4).

Comment #2.

I am curious about the case with the BRAF gene fusion. BRAF fusions have been reported in myxoinflammatory fibroblastic sarcoma, which can mimic MFS. Do the authors think that this is a mis-classified case, or, instead, another example of shared molecular features (such as amplification of VGLL3 in a subset of both entities)?

We appreciate your comment. In the original pathologic review process, the distinction between MFS and MIFS was carefully made, and we confirmed that no MIFS inadvertently contaminated the study cohort. According to your suggestion, we re-reviewed medical record and glass slides of MFS51 with *SLC37A3-BRAF* fusion; however, the tumor

displayed no histological features suggestive of MIFS. In addition, this case had TP53 and ATRX mutations. We therefore believe that *BRAF* fusion is just a shared abnormality between 2 different entities of sarcoma that have histological similarity. We have added the following statement in the Discussion section.

Page 21, Line 18

“Although *BRAF* fusion has been recently identified in a subset of myxoinflammatory fibroblastic sarcoma⁵⁶, our MFS with *BRAF* fusion showed no findings that may suggest relationship with this rare low-grade acral sarcoma.”

Comment #3.

In the up- and down-regulated gene list – what is the basis for comparison? (median of all genes?)

We appreciate your comments. To identify up- and down-regulated genes, we applied Welch's t-test to every genes by comparing the two groups; six higher expressing samples (20%) and others (80%) for identifying up-regulated genes and six lower expressing samples (20%) and others (80%) for identifying down-regulated genes. We described the method for identifying up-regulated genes in the original version of the manuscript, therefore, we added the method for identifying down-regulated genes in the revised version of the manuscript (Page 28, Line 16).

Comment #4.

In the TCGA data, most sarcoma types with hypermethylation had worse outcomes than hypomethylated tumors – I would be curious as to what the authors speculate could account for their findings that the opposite was true in their series of MFS?

We appreciate your suggestive comment. As the reviewer pointed out, the TCGA paper reported that hypermethylated cases had worse outcomes than hypomethylated cases in dedifferentiated liposarcoma and soft tissue leiomyosarcoma. However, we are not able to find any description about the relation between methylation status and clinical outcome of MFS/UPS in the TCGA paper, and we think that the relationship in MFS remains to be determined in that report.

Hypomethylation, as well as hypermethylation, of CpG islands was reported to be associated with worse clinical outcome depending on the cancer types (Kusano M et al, Cancer 2006;106(7):1467-1479). One possible reason why hypomethylation was a poor prognostic feature in MFS could be that MFS is biologically or genealogically distinct and may have acquired a different methylation profile during progression from those of dedifferentiated liposarcoma and soft tissue leiomyosarcoma

In addition, genome-wide DNA hypomethylation, that leads to the activation of retrotransposon, such as increased LINE-1 expression levels, and genomic instability, was also known to be associated with poor prognostic outcome in a range of tumor types (Xiao-Jie L et al, Genet Med 2016;18(5):431-439). Therefore, we can also speculate that genome-wide DNA hypomethylation may play a more influential role in the progression of MFS compared to other soft tissue sarcomas that more depend on hypermethylation of CpG islands.

In any cases, as our observation was based on relatively small number of cases, further investigation with a greater number of samples are required to draw a more conclusive statement regarding the clinical implication of methylation profile in MFS.

Comment #5.

Some of the supplemental tables have different numbering on the excel tab than the file was submitted as (i.e table S13 the excel tab labels it as S12 though the heading says S13).

Comment #6.

Also consider for simplicity's sake submitting several of the related tables as one excel file with tabs (assuming journal guidelines allow) – i.e Tables S10-13 could be tabs A-D in one table.

We appreciate your comments for the supplementary tables. We complied and renumbered the supplementary tables as suggested.

Comment #7.

As before, I am unclear on why the authors chose to reiterate the same data shown in figure 1 as 2a, 3a and-4a – It isn't that hard to follow; I would rather see these removed, condense the pathway panels into one figure and some of the more interesting findings from the supplemental data moved to main figures (such as the immune analysis for the methylation clusters). But I agree this may be a personal preference of mine, and others may feel differently.

We appreciate your comment. We reformatted and renumbered figures and supplementary figures as suggested. According to this change, figure legends were modified (Page 37-39).

Comment #8.

I also wonder why the authors chose only to mention one rare fusion (albeit one involving BRAF), when there are several other recurrent fusions listed in the supplemental table that might be of interest despite not involving a known oncogene. While I acknowledge that generating cell models such as was done for the BRAF fusion for these fusions are beyond the scope of this paper, It would have been nice to include some speculation about these recurrent fusions and their functionality, if any.

We appreciate your comment. We totally agree with your comment and added more discussions regarding other recurrent and potentially important fusion genes identified in our cohort such as those involving *TRIO*, which were reported in 2017 by a French group (Reference 57) and also TCGA sarcoma group (Reference 10). We modified the Discussion section regarding fusion genes identified in this study by adding the statement of *TRIO* fusion gene as below (Page 22, Line 3).

“ Our RNA sequencing identified 71 in-frame recurrent fusion genes as well as many singleton ones in 29 MFSs. They included a *TRIO* gene fusion that was reported previously⁵⁷. In this report, RNA sequencing of 117 non-translocation-related sarcomas, including 15 MFSs, identified one recurrent *TRIO* fusion genes with various partners in 7

cases including one MFS. *TRIO* fusion gene was also found in one of 17 MFSs in the TCGA data¹⁰. Although *TRIO* fusion genes were speculated to be associated with a transcriptomic program of immunity/inflammation, cell proliferation and migration, their oncogenic roles have been still unclear. Further functional analysis of this recurrent gene fusion would permit to uncover therapeutic applications for MFS with *TRIO* rearrangements.”

Reviewer #2 (Remarks to the Author):

I thank the authors for taking such care to address all of my concerns and additional suggestions, despite our not seeing eye-to-eye on the MDM2 amplification controversy. I find the revised figures much more helpful to illustrate the text than the earlier version. Thank you!